# GUIDED STAR-SHAPED MASKED DIFFUSION

## ABSTRACT

The performance of pre-trained masked diffusion models is often constrained by their sampling procedure, which makes decisions irreversible and struggles in low-step generation regimes. We introduce a novel sampling algorithm that works with pre-trained models and, after a lightweight fine-tuning of a single layer, significantly improves sample quality and efficiency. Our method reformulates the generation process using a star-shaped paradigm, which inherently allows for error correction. To make this process effective, we augment it with a learnable re-masking scheduler that intelligently identifies and revises likely errors. This approach yields a substantial quality boost, particularly when using a small number of sampling steps. We extensively ablate key components of our approach and show its usability in different scenarios. In comprehensive experiments on text, and code generation, our sampling algorithm outperforms or matches existing methods.

## 1 INTRODUCTION

Diffusion probabilistic models have demonstrated remarkable success in generating high-fidelity data, particularly in continuous domains such as image and video synthesis (Sohl-Dickstein et al., 2015; Song & Ermon, 2019; Ho et al., 2020; Sahoo et al., 2024b). A key reason for their effectiveness is the principle of iterative refinement. By progressively denoising a sample from a simple prior distribution, these models effectively sculpt data, making small adjustments at each step. This allows for a robust error correction mechanism; a mistake made early in the trajectory can be gradually amended in subsequent steps, leading to state-of-the-art results.

This elegant property, however, is largely absent in the discrete domain. While discrete diffusion models are making significant strides in areas like natural language processing (Lou et al., 2024; Sahoo et al., 2024a; Schiff et al., 2024), the most successful variants, based on token masking, are built on a foundation that precludes iterative refinement. In a masked diffusion setup, the generation of each token is a one-way street: once a [MASK] is replaced with a concrete token, the model commits to that decision. The token is then frozen and cannot be revisited or updated, even if later steps reveal it to be suboptimal in the broader context. This sequence of irreversible commitments prevents the model from correcting its own mistakes, imposing a fundamental ceiling on sample quality, sampling speed, and the potential for fine-grained, controlled generation.

Recognizing this limitation, several recent works have proposed mechanisms to enable token revision. For instance, ReMDM (Wang et al.) introduces a simple yet effective strategy: randomly re-masking a fraction of already-generated tokens during the sampling process. While this approach yields substantial quality improvements in text generation, its stochastic nature is fundamentally inefficient. The selection process is indiscriminate; it is just as likely to re-mask a correct token as an erroneous one, unnecessarily slowing convergence and requiring a large number of sampling steps. An alternative approach, explored by GIDD (von Rütte et al., 2025), combines masked diffusion with a uniform diffusion process to allow for token refinement toward the end of generation. However, this hybrid strategy has not achieved yet competitive sample quality. These pioneering efforts highlight the need for an error correction mechanism, yet they also reveal the limitations of non-selective revision, motivating our targeted approach.

To address these shortcomings, we propose a new sampling framework founded on the star-shaped paradigm (Okhotin et al., 2023). Instead of a direct, irreversible step from state $\mathbf{x}_t$ to $\mathbf{x}_s$, our sampler first predicts a complete version of the clean data, $\hat{\mathbf{x}}_0 \sim p_\theta(\cdot \mid \mathbf{x}_t)$, and then gener-

ates the next, less noisy state by sampling from the forward process conditional on this prediction, $\mathbf{x}_s \sim q(\cdot \mid \hat{\mathbf{x}}_0)$. This two-step process inherently breaks the chain of immutable decisions, allowing already-generated tokens to be re-masked and refined. Crucially, this formulation is compatible with pre-trained Masked Diffusion Language Models (MDLMs), allowing us to enhance existing pretrained models with a new sampling procedure.

To unlock the full potential of the reversible star-shaped sampler, we replace its inefficient unguided remasking with a lightweight, learnable module trained to target tokens predicted to be erroneous. The resulting method, Guided Star-Shaped Masked Diffusion (G-Star), yields a substantial quality boost, particularly in computationally constrained, few-step generation regimes.

Our main contributions are threefold:

- We propose a star-shaped formulation for masked discrete diffusion that enables iterative refinement and error correction.
- We introduce a learned masking scheduler that adaptively identifies and remasks tokens predicted to be erroneous. This intelligent error targeting mechanism significantly accelerates inference and improves final sample quality.
- We demonstrate empirically that our approach achieves superior sampling performance across a diverse set of domains, including text and code generation.

## 2 PRELIMINARIES

Our work builds upon masked diffusion models, which operate by progressively masking and unmasking tokens.

**Masked diffusion models.** We consider discrete tokens represented as one-hot vectors $\mathbf{x} \in \{0,1\}^{|V|}$, where $|V|$ is the vocabulary size. A special [MASK] token is denoted by $\mathbf{m}$. The forward process corrupts an input $\mathbf{x}_0$ by progressively masking tokens over $T$ timesteps according to a noise schedule $\alpha_t$. The marginal distribution of the noisy state $\mathbf{x}_t$ is given by:

$$q(\mathbf{x}_t \mid \mathbf{x}_0) = \text{Cat}(\mathbf{x}_t; \alpha_t \mathbf{x}_0 + (1 - \alpha_t)\mathbf{m}). \tag{1}$$

The reverse process is parameterized by a neural network, $f_\theta(\mathbf{x}_t, t)$, which is trained to predict the probability distribution over the original data, $p_\theta(\mathbf{x}_0 \mid \mathbf{x}_t)$. This predicted distribution then conditions the analytical posterior:

$$q(\mathbf{x}_{t-1} \mid \mathbf{x}_t, \mathbf{x}_0) = \begin{cases} \delta_{\mathbf{x}_t}(\mathbf{x}_{t-1}), & \text{if } \mathbf{x}_t \neq \mathbf{m} \\ \text{Cat}\left(\mathbf{x}_{t-1}; \frac{(1-\alpha_{t-1})\mathbf{m} + (\alpha_{t-1} - \alpha_t)\mathbf{x}_0}{1 - \alpha_t}\right), & \text{if } \mathbf{x}_t = \mathbf{m} \end{cases} \tag{2}$$

The first case of this posterior, where an unmasked token is deterministically preserved ($\delta_{\mathbf{x}_t}$), reveals the model's core limitation: once a token is generated, it is frozen, making iterative error correction impossible. The network $f_\theta$ is typically trained by minimizing a weighted cross-entropy loss to predict $\mathbf{x}_0$ from $\mathbf{x}_t$.

**ReMasking diffusion models.** To address this limitation, ReMDM (Wang et al.) introduces new sampling process that allows already-unmasked tokens to be reverted to a [MASK] state. This is achieved by modifying the posterior:

$$q(\mathbf{x}_{t-1} \mid \mathbf{x}_t, \mathbf{x}_0) = \begin{cases} \text{Cat}(\mathbf{x}_{t-1}; (1 - \sigma_t)\mathbf{x}_0 + \sigma_t \mathbf{m}), & \text{if } \mathbf{x}_t \neq \mathbf{m} \\ \text{Cat}\left(\mathbf{x}_{t-1}; \frac{\alpha_{t-1} - (1-\sigma_t)\alpha_t}{1 - \alpha_t}\mathbf{x}_0 + \frac{1 - \alpha_{t-1} - \sigma_t \alpha_t}{1 - \alpha_t}\mathbf{m}\right), & \text{if } \mathbf{x}_t = \mathbf{m} \end{cases} \tag{3}$$

Here, the hyperparameter $\sigma_t \in [0; min\{1, \frac{1-\alpha_{t-1}}{\alpha_t}\}]$ controls the re-masking probability for already unmasked tokens. A key practical challenge of this method is that $\sigma_t$ is determined by one of several proposed schedules, each governed by a hyperparameter $\eta$ that must be carefully tuned (see Appendix C.1). While this enables error correction that is compatible with pre-trained models, its effectiveness is limited by the non-selective nature of the remasking schedule, motivating a more targeted approach.

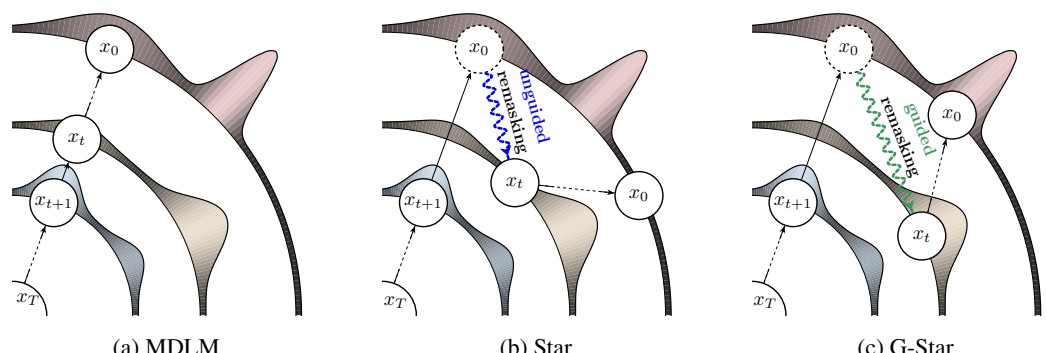

(a) MDLM         (b) Star         (c) G-Star

Figure 1: Comparison of three sampling trajectories for refining text from a noisy state ($x_T$) to a clean state ($x_0$). The orbits represent the probability of partially denoised text $x_t$ at each step. **(a) MDLM** follows a one-way, step-by-step path; it is stable but unable to correct past mistakes. **(b) Star sampler** allows revision by predicting $x_0$ and then **randomly re-masking** tokens, regardless of whether they are correct or incorrect. This allows for correction but is suboptimal and can harm text coherence. **(c) G-Star sampler** is an improved path that also predicts $x_0$ but uses an error predictor to **selectively re-mask** likely incorrect tokens, enabling efficient error correction while preserving text quality.

## 3 GUIDED STAR-SHAPED MASKED DIFFUSION

The fundamental limitation of standard masked diffusion, as outlined in Section 2, is its irreversible structure. To enable iterative refinement, we must break this chain of immutable decisions. In this section, we introduce a star-shaped paradigm for masked diffusion that allows for token revision, and then present a learned scheduler that makes this revision process efficient and targeted.

### 3.1 STAR-SHAPED MASKED DIFFUSION

We redefine the joint distribution of the forward process. Instead of conditioning each latent state $\mathbf{x}_t$ on its immediate predecessor $\mathbf{x}_{t-1}$, we make all latent states conditionally independent given the original data $\mathbf{x}_0$:

$$q(\mathbf{x}_{1:T} \mid \mathbf{x}_0) = \prod_{t=1}^{T} q(\mathbf{x}_t \mid \mathbf{x}_0). \tag{4}$$

Here, each $q(\mathbf{x}_t \mid \mathbf{x}_0)$ is the same marginal distribution defined in Equation 1. This "star-shaped" structure, where all latents connect directly to $\mathbf{x}_0$, fundamentally alters the process dynamics. It explicitly permits non-monotonic transitions; for instance, a token can be masked at a timestep $s$ and become unmasked at a later timestep $t > s$, a scenario forbidden in the standard Markovian chain for masked diffusion.

This change simplifies the reverse posterior: $q(\mathbf{x}_{t-1} \mid \mathbf{x}_t, \mathbf{x}_0) = q(\mathbf{x}_{t-1} \mid \mathbf{x}_0)$. Following the standard diffusion paradigm, we construct the generative transition $p_\theta(\mathbf{x}_{t-1} \mid \mathbf{x}_t)$ by first predicting an estimate of the clean data, $\hat{\mathbf{x}}_0 \sim \mathrm{Cat}(\cdot, f_\theta(\mathbf{x}_t, t))$, and then sampling from the corresponding posterior:

$$p_\theta(\mathbf{x}_{t-1} \mid \mathbf{x}_t) = q(\mathbf{x}_{t-1} \mid \mathbf{x}_0 = \hat{\mathbf{x}}_0). \tag{5}$$

Intuitively, each step of the generative process involves two stages: (1) the model examines the current state $\mathbf{x}_t$ and forms a complete hypothesis about the final, clean data $\hat{\mathbf{x}}_0$; (2) it then generates the next, less noisy state $\mathbf{x}_{t-1}$ by applying the forward noising process to this hypothesis, effectively remasking it to the appropriate noise level. This step is what allows the model to revise its previous decisions (see Figure 1b).

Notably, this star-shaped sampling process establishes a direct connection to the ReMDM framework (Wang et al.). Specifically, our sampler is mathematically equivalent to the ReMDM sampler when its probability is set to $\sigma_t = 1 - \alpha_s$. As we demonstrate in our analysis (Appendix C.1 and Section 4.4), this hyperparameter requires an extensive, per-schedule tuning process to be effective.

**Algorithm 1** Training the error predictor $g_\phi$

1: **Input:** Dataset $\mathcal{D}$, pre-trained diffusion model $f_\theta$, learning rate $\eta$, denoiser temperature $\tau_{\text{denoiser}}$.
2: **Output:** Trained error predictor $g_\phi$
3: **while** not converged **do**
4:      Sample batch $\{\mathbf{x}_0\} \sim \mathcal{D}$
5:      ▷ Simulate denoising and identify errors $y$
6:      $t \sim \mathcal{U}(0,1)$
7:      $\mathbf{x}_t \sim q(\cdot \mid \mathbf{x}_0)$
8:      $\hat{p}_0 \leftarrow \text{Softmax}(\frac{f_\theta(\mathbf{x}_t)}{\tau_{\text{denoiser}}})$
9:      $\hat{\mathbf{x}}_0 \sim \text{Cat}(\cdot; \hat{p}_0)$
10:     $y \in \{0,1\}^L$, where $y_i = \mathbb{I}(\hat{\mathbf{x}}_{0,i} \neq \mathbf{x}_{0,i})$
11:     ▷ Train the error predictor
12:     $p \leftarrow \text{Softmax}(g_\phi(\hat{\mathbf{x}}_0))$
13:     $\mathcal{L}_\phi \leftarrow -\frac{1}{L}\sum_{i=1}^{L} \big[\, y_i \log p_i$
                               $+ (1 - y_i)\log(1 - p_i)\big]$
14:     $\phi \leftarrow \phi - \eta \nabla_\phi \mathcal{L}_\phi$
15: **return** $g_\phi$

**Algorithm 2** Guided sampling step

1: **Input:** Current state $\mathbf{x}_t$, current time $t$, diffusion model $f_\theta$, error predictor $g_\phi$, denoiser temperature $\tau_{\text{denoiser}}$, nucleus probability $p_{\text{nucleus}}$, error predictor temperature $\tau_{\text{remask}}$
2: **Output:** Next state $\mathbf{x}_{t-1}$
3: ▷ Predict and sample a proposal clean state
4: $\hat{p}_0 \leftarrow \text{NucleusFilter}(\text{Softmax}(\frac{f_\theta(\mathbf{x}_t)}{\tau_{\text{denoiser}}}), p_{\text{nucleus}})$
5: $\hat{\mathbf{x}}_0 \sim \text{Cat}(\cdot; \hat{p}_0)$
6: ▷ Identify and select most likely errors
7: $\text{logits}_{\text{err}} \leftarrow g_\phi(\hat{\mathbf{x}}_0)$
8: $N \leftarrow \lceil (1 - \alpha_{t-1}) \cdot L \rceil$
9: $\mathcal{M} \leftarrow \text{SampleKNoRep}(\frac{\text{logits}_{\text{err}}}{\tau_{\text{remask}}}, N)$
10: ▷ Construct next state via targeted remasking
11: $\mathbf{x}_{t-1,i} \leftarrow \begin{cases} \mathbf{m}, & \text{if } i \in \mathcal{M} \\ \hat{\mathbf{x}}_{0,i}, & \text{otherwise} \end{cases}$
12: **return** $\mathbf{x}_{t-1}$

Our formulation avoids this costly search and allows our sampler to perform on par with a carefully optimized ReMDM.

A crucial consequence of this formulation is its compatibility with existing models. The variational lower bound (VLB) for this process can be simplified to a weighted cross-entropy objective, structurally identical to that used for standard masked diffusion models:

$$\mathcal{L} \approx \mathbb{E}_{t,\mathbf{x}_0,\mathbf{x}_t}\left[-w'_t \log p_\theta(\mathbf{x}_0 \mid \mathbf{x}_t)\right]. \tag{6}$$

**Claim 1.** *The VLB for the star-shaped process simplifies to the objective in Eq. equation 6, which has the same functional form as the standard masked diffusion objective but with different timestep-dependent weights $w'_t$.* (Proof in Appendix A).

The structural similarity between our training objective and the standard MDLM loss motivates the reuse of pre-trained MDLM weights as an effective practical strategy. We empirically confirm this approach, finding that it allows our sampler to achieve strong performance without any fine-tuning.

### 3.2 LEARNED ERROR-TARGETED

While the sampler described in Equation 5 enables error correction, it is inefficient. The remasking process is non-selective — it samples from $q(\mathbf{x}_{t-1} \mid \hat{\mathbf{x}}_0)$, which is just as likely to mask a correct token as an incorrect one. This negatively impacts both sampling efficiency and final sample quality.

To rectify this, we introduce a secondary model: an error predictor $g_\phi$, which learns to identify which tokens the primary diffusion model $f_\theta$ is likely to get wrong. This allows us to focus the procedure on probable errors.

**Training the error predictor.** The purpose of the error predictor, $g_\phi$, is to learn to identify which tokens the main diffusion model, $f_\theta$, is likely to generate incorrectly. To train it, we simulate this error-making process. First, we take a clean text from the training data and apply the forward diffusion process to corrupt it with [MASK] tokens, creating a noisy input that mimics a state during generation. Next, we feed this masked text to our pre-trained diffusion model, $f_\theta$, which predicts a probability distribution over the clean text. By sampling from this distribution, we obtain a discrete candidate sequence. This candidate will inevitably contain some errors where the model's prediction does not match the ground truth. The error predictor's task is to learn to spot these mistakes: it is trained to take the candidate sequence as input and predict which of its tokens are incorrect. The entire procedure is detailed in Algorithm 1.

**Inference with targeted.** During generation, we incorporate the trained error predictor $g_\phi$ to guide the remasking process, replacing the sampler's indiscriminate selection of tokens with a targeted procedure (see Figure 1c). The process for each sampling step from $\mathbf{x}_t$ to $\mathbf{x}_s$, detailed in Algorithm 2, proceeds as follows. First, the main diffusion model $f_\theta$ generates a clean data candidate,

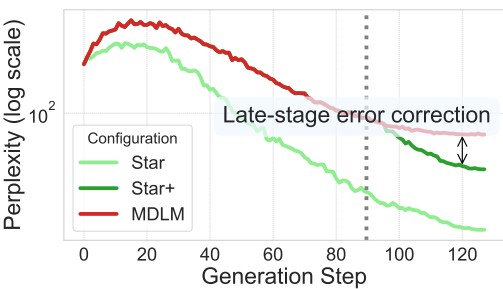 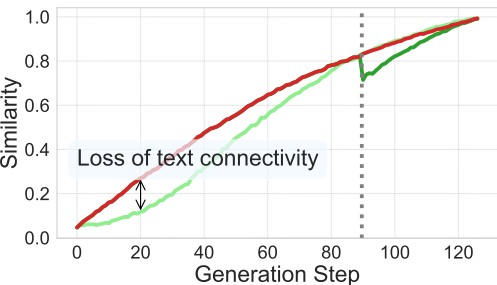

Figure 2: Analysis of the star-shaped (Star) sampler's dynamics. *(Left)* Perplexity and *(Right)* step-to-step similarity over the generation trajectory for three configurations: MDLM, Star, and our hybrid approach (Star+), which switches from MDLM to Star at step 90 (dotted line).

$\hat{\mathbf{x}}_0$. This candidate is then scored by the error predictor $g_\phi$ to obtain error logits for each token. These logits, scaled by a temperature $\tau_{\text{remask}}$, are then used to sample the $N$ locations without repetitions via Gumbel-Top-K trick sampling (Kool et al., 2019), where $N$ is determined by the noise schedule. The next state $\mathbf{x}_s$ is then formed by reverting these targeted tokens in $\hat{\mathbf{x}}_0$ back to the [MASK] symbol. This targeted approach focuses the model's capacity on correcting its most probable mistakes, thereby improving sampling efficiency and final quality. As we will demonstrate in our analysis (Sections 4 and 5), integrating this targeted remasking mechanism allows for a significant improvement in generation quality at the cost of only a minor increase in parameter overhead.

## 4 ANALYSIS

This section deconstructs our proposed sampling method and validates its key components through a series of controlled experiments. We analyze: (1) the optimal scheduling for the star-shaped sampler, identifying the distinct generative phases where it is most beneficial; (2) the contribution of the guidance mechanism to improving sample quality and step efficiency; (3) the sampler's performance within the iterative refinement context of the ReMDM loop-schedule protocol; and (4) the architectural requirements of the error predictor, confirming the efficacy of a highly parameter-efficient design.

### 4.1 EXPERIMENTAL SETUP

All analytical experiments are conducted on the OpenWebText (OWT) dataset (Gokaslan & Cohen, 2019), tokenized using the standard `gpt-2` tokenizer (Radford et al., 2019). For these experiments, we fine-tuned the publicly available MDLM checkpoint from Sahoo et al. (2024a) for unconditional generation of 128 and 512-token sequences, padding shorter outputs where necessary. We generate 5,000 samples for each configuration and assess performance using a suite of three complementary metrics. Sample quality and local coherence are measured via Perplexity (PPL), computed using a pre-trained GPT-2 LARGE model (Radford et al., 2019). Lexical variety is quantified by the Diversity (DIV) score, defined as $\text{div}(y) = \prod_{n=2}^{4} \frac{\# \text{ unique } n\text{-grams in } y}{\# \, n\text{-grams in } y}$. Finally, to provide a more holistic assessment that balances quality with diversity, we report the MAUVE score (Pillutla et al., 2021), which measures the distributional alignment between the generated and reference texts.

### 4.2 WHEN TO USE THE STAR-SHAPED SAMPLER?

Our initial experiments revealed a critical insight: the pure star-shaped (Star) sampler, when applied across the entire generation trajectory, exhibits poor performance and often leads to degenerate text. This observation motivated our central hypothesis: the generation process is not monolithic but consists of two distinct phases, each benefiting from a different sampling strategy. We posit that the initial phase requires a stable, structure-building sampler, while the final phase benefits from an error-correcting one. To test this hypothesis, we first analyze this phenomenon in a simplified setting involving the generation of 128-token sequences from the OWT dataset.

**Phase 1: the challenge of early-stage generation.** In the early stages of generation (high $t$), a large fraction of tokens is masked. The star-shaped sampler's strategy of predicting a full $\hat{\mathbf{x}}_0$ requires the model to generate a large number of new tokens conditioned on a very sparse context. While these newly generated tokens may be individually plausible with respect to the unmasked context, they often lack mutual coherence among themselves. The problem is exacerbated by the subsequent step of the star-shaped process: the independent, random remasking of all tokens in this new hypothesis. This process may preserve a large fraction of the newly generated, yet mutually incoherent, tokens while masking others that provided the original context. As a result, the input for the next iteration becomes an increasingly fragmented and incoherent context. This complicates the subsequent prediction task, causing errors to compound over iterations and ultimately leading to the observed text degradation.

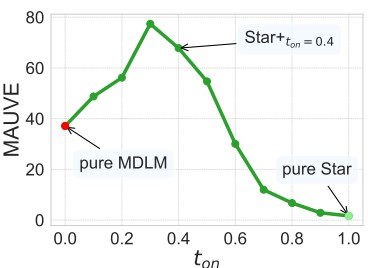

Figure 3: The impact of the star-shaped sampler's activation time ($t_{on}$) on generation quality. We plot the final MAUVE score for a hybrid sampler that switches from MDLM to Star at time $t_{on}$.

This generative incoherence is empirically captured in Figure 2 (right), where the pure star-shaped (Star) sampler (light green line) demonstrates significantly lower step-to-step similarity than the standard MDLM (red line). This metric, defined as the fraction of matching tokens in the predicted clean data ($\hat{\mathbf{x}}_0$) between adjacent steps,. The low score for Star sampler confirms that its generation process struggles to build upon a coherent structure. This ultimately leads to text degradation, as reflected by its near-zero MAUVE score (see Figure 3 at $t_{on} = 1.0$). In contrast, the MDLM's incremental, token-by-token generation ensures high step-to-step similarity, allowing it to stably construct a coherent draft.

**Phase 2: the power of late-stage refinement.** While the MDLM's stability is advantageous for initial structure-building, its irreversible nature limits its ability to correct errors. This is where the star-shaped paradigm excels. In the late stages of generation (low $t$), the vast majority of tokens are already determined, providing a strong, coherent conditioning context. Remasking a small fraction of these tokens and repredicting them from a global perspective $\hat{\mathbf{x}}_0$ becomes a powerful mechanism for error correction, rather than a source of instability.

This effect is visible in Figure 2 (left). When our hybrid Star+ sampler switches from MDLM to star-shaped sampler at step 90 (dotted line), its perplexity (green line) begins to decrease more rapidly than the pure MDLM baseline, ultimately achieving a superior final score. This demonstrates that the Star sampler is highly effective at refining an already well-formed text.

**Empirical validation: finding the optimal transition point.** To validate this two-phase hypothesis and identify the optimal transition point, we conduct an ablation study on the activation time, $t_{on}$. The sampler operates as a standard MDLM until time $t_{on}$, after which it switches to the star-shaped paradigm. Figure 3 plots the final MAUVE score as a function of $t_{on}$. The results provide strong empirical support for our hypothesis. Performance is poor for both pure samplers ($t_{on} = 1.0$ for pure Star and $t_{on} = 0.0$ for pure MDLM) but peaks at $t_{on} \approx 0.3$. This confirms that the most effective strategy is to leverage the MDLM process for the initial $60 - 80\%$ of the generation to build a coherent draft, and then activate the star-shaped sampler for the final $20 - 40\%$ for global refinement.

### 4.3 GUIDED STAR-SHAPED SAMPLER

The preceding analysis established that a hybrid sampler (Star+) effectively refines text in the late stages of generation. However, its reliance on unguided remasking is inherently sample-inefficient. This raises a central question: can we significantly improve performance by replacing this stochastic process with a targeted, intelligent one? In this section, we test this hypothesis by introducing our full proposed method, the **Guided Star-shaped sampler (G-Star)**, which uses an error predictor to focus the refinement process exclusively on likely errors. We posit that the primary advantage of this targeted approach will manifest in computationally constrained, few-step generation regimes, where the efficiency of each correction step is paramount.

To validate this, we perform a direct comparison between the unguided Star+ and our guided G-Star+ (both employ the identical hybrid switching schedule, $t_{on} = 0.2$) sampler on the task of generating 512-token sequences from OpenWebText, evaluating across a range of sampling step counts from 32 to 512. The results are presented in Figure 4.

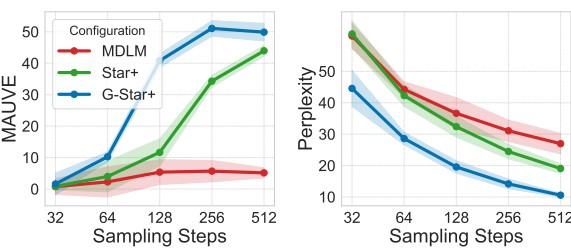

The empirical evidence strongly supports our hypothesis. The MAUVE scores (left panel) show that while both hybrid samplers outperform the MDLM baseline, the

Figure 4: Performance comparison in few-step generation regimes. Guided sampler (G-Star+) consistently outperforms the unguided Star+.

guided G-Star+ variant achieves significantly higher distributional fidelity. Crucially, the performance gap is most pronounced in the medium-step regimes of 64-256 steps. While at the 32-step mark all samplers struggle, our guided approach still shows a modest advantage. This gap then narrows as the step budget increases.

This dynamic has a clear and intuitive explanation. With a large number of sampling steps, even an unguided remasking process has a high probability of eventually correcting most errors, causing the performance of the two samplers to converge. However, when each step is critical, the intelligent targeting provided by the error predictor becomes the deciding factor. By focusing the model's capacity on the most probable errors, the guidance mechanism ensures that each refinement step is maximally impactful. This enables the generation of higher-quality text with a significantly reduced computational budget, highlighting the practical advantage of our guided approach. A qualitative visualization of the refinement process for both samplers is available in Appendix F.

## 4.4 ITERATIVE REFINEMENT REGIME

To further analyze the refinement capabilities of our sampler, we adopt the **loop schedule** protocol introduced by ReMDM (Wang et al.). This specialized schedule is designed to evaluate a sampler's efficiency at refining an already generated text. The process consists of three distinct phases: (1) an initial generation phase using the standard MDLM sampler to produce a coherent draft; (2) a refinement phase, where a fixed number of "looping" steps are performed at a constant noise level ($\alpha_t = 0.9$) to iteratively improve the draft; and (3) a final generation phase to complete the sequence.

We implement this protocol for the task of generating 512-token sequences from OpenWebText and compare our unguided (Star-loop) and guided (G-Star-loop) samplers. We test three configurations with varying computational budgets, corresponding to a total of 128, 256, and 512 generation steps. It is important to note that achieving these strong ReMDM results requires an extensive, per-schedule search for the hyperparameter $\eta$, as detailed in Appendix C.1. This tuning process is a significant drawback, as performance can vary from very strong to worse than the baseline MDLM. In contrast, our unguided Star-loop sampler performs competitively without requiring any tuning of $\eta$, highlighting a key practical advantage of the proposed star-shaped formulation.

Beyond scheduling, we also observe that the **denoiser's temperature** ($\tau_{\text{denoiser}}$) provides an additional and complementary axis of control over the generation process. By adjusting the softmax temperature applied to the denoiser's output logits, the sampler can smoothly trade off between perplexity and diversity: lower temperatures make the denoiser more deterministic, tightening the distribution around high-confidence tokens and leading to lower perplexity but reduced lexical variety, while higher temperatures increase stochasticity in the predictions, yielding more diverse samples at a cost in quality. In practice, adjusting the denoiser temperature thus provides a lightweight and effective control knob for navigating the perplexity–diversity frontier, enabling a more thorough examination and comparison of model behavior.

The Pareto fronts in Fig. 5a highlight the clear advantage of our guided refinement strategy. Across all computational budgets, G-Star-loop consistently achieves a superior balance between perplexity and diversity, outperforming both the MDLM baseline and the Star/ReMDM-loop variants. Remarkably, even in the severely constrained 128-step regime, G-Star-loop attains substantially lower perplexity while simultaneously delivering higher diversity than the best-tuned ReMDM configuration

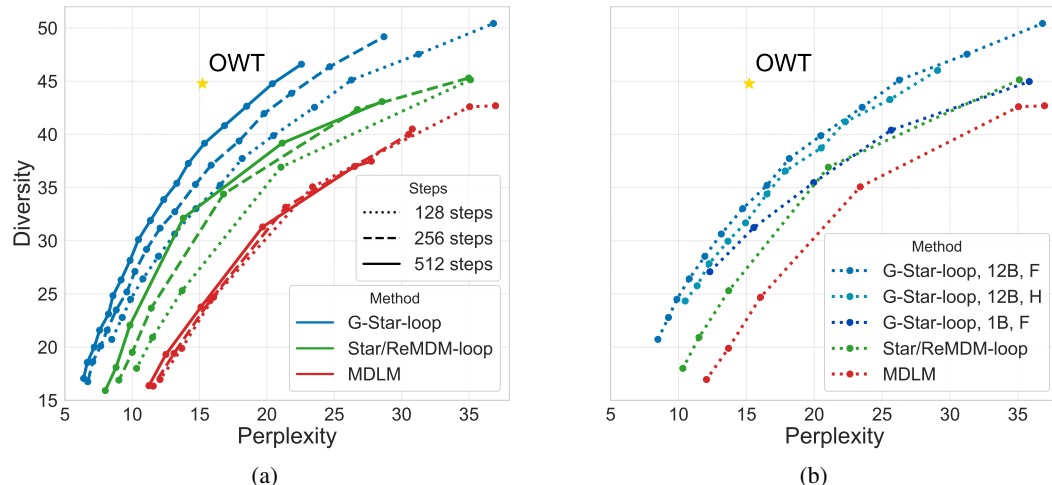

(a)                  (b)

Figure 5: Pareto fronts for different methods obtained by varying the denoiser temperature. The left plot compares MDLM, Star/ReMDM-loop, and G-Star-loop across different sampling steps, while the right plot compares different G-Star configurations (12B–F, 12B–H, 1B–F) against Star/ReMDM-loop and MDLM for 128 denoising steps.

with 512 steps. These findings confirm that targeted refinement, rather than stochastic remasking, is essential for efficient quality improvements. Additional results you can find in Appendix C.4.

### 4.5 ERROR PREDICTOR CAPACITY AND EFFICIENCY

We investigate the trade-off between the error predictor's model capacity and its performance. To this end, we evaluate three architectural configurations initialized from the pre-trained MDLM: a lightweight model using one (`1B, F`) transformer block with full fine-tuning, and full 12-block models with either all weights fine-tuned (`12B, F`) or only the classification head trained (`12B, H`). Using the same Pareto-front evaluation setup as before, but restricting the sampler to the 128-step regime, the results in Fig. 5b lead to two main conclusions. First, the parameter-efficient head-only variant (`12B, H`) closely tracks the performance of the fully fine-tuned model (`12B, F`) across the entire frontier, indicating that the pre-trained MDLM representations already provide strong features for error prediction. Second, the lightweight `1B, F` predictor is clearly less competitive: it only outperforms the MDLM and Star/ReMDM-loop baselines in the low-diversity, low-temperature corner of the frontier and matches their performance elsewhere without providing additional gains. Overall, this suggests that while head-only training offers an excellent efficiency–quality trade-off, more aggressive capacity reduction can noticeably degrade refinement performance.

## 5 EMPIRICAL EVALUATION

Having analyzed the internal mechanics and key components of our sampler on the OWT dataset, we now turn to evaluating its performance and general applicability in a broader context. In this section, we benchmark G-Star on two challenging generative tasks: (1) **large-scale language modeling**, where we assess performance on downstream benchmarks to validate its effectiveness at scale, and (2) **source code generation** on the Conala benchmark (Yin et al., 2018).

### 5.1 APPLICATION TO LARGE-SCALE INSTRUCTION-TUNED MODEL

In this section we investigate whether our guided sampler can enhance the performance of an instruction-tuned large language model. For this purpose, we integrate our G-Star sampler into the **Dream-Instruct 7B** (Ye et al., 2025b) model and evaluate it on a diverse suite of complex downstream benchmarks. We establish our baseline by evaluating the Dream-Instruct model with the authors' official configuration. As shown in Table 1, our reproduced scores vary slightly from the originally published results and serve as the direct point of comparison for our method.

Table 1: Downstream benchmark results for Dream-Instruct 7B. The best result is marked in **bold**.

|  | Dream-Ins. (Paper) | Dream-Ins. (Reproduced) | + G-Star (Ours) |
|---|---|---|---|
| MMLU | 67.0 | 69.9 | **71.2** |
| MMLU-PRO | 43.3 | 46.9 | **47.9** |
| GSM8K | 81.0 | 81.5 | **81.6** |
| GPQA | 33.0 | 31.0 | **32.8** |
| HumanEval | 55.5 | 53.7 | 54.9 |
| MBPP | 58.8 | 58.0 | **59.4** |
| IFEval | 62.5 | 56.4 | **59.3** |

Table 2: Conditional perplexity on the Conala benchmark for different samplers and step counts. **Best** and second-best results are highlighted.

| Algorithm | Qwen2.5B-Coder ppl ↓ | | |
|---|---|---|---|
|  | 32 steps | 64 steps | 128 steps |
| MDLM | 29.8 | 25.5 | 26.7 |
| ReMDM-loop$_{\eta=0.02}$ | 30.1 | 25.0 | 20.4 |
| ReMDM-cap$_{\eta=0.04}$ | 27.3 | 22.5 | 19.1 |
| G-Star-loop | 22.5 | **17.8** | 17.8 |
| G-Star+$_{t_{on}=0.3}$ | **20.4** | 18.9 | **16.4** |

For our approach, we augment the Dream-Instruct baseline with our G-Star sampler, integrating it via a loop-based refinement strategy. We keep the total number of diffusion steps identical to the baseline but designate 10% of them as refinement steps executed by G-Star at a specific noise level $\alpha_{on}$. The error predictor is configured for maximum parameter efficiency: we freeze the 7B model's backbone and train only a lightweight classification head on the Tulu 3 (Lambert et al., 2024) dataset. Detailed configurations for each benchmark are provided in Appendix E.

As summarized in Table 1, our G-Star sampler yields consistent performance gains across all seven evaluated benchmarks. We observe noteworthy improvements on complex reasoning tasks such as MMLU (+1.3 points) and GPQA (+1.8 points), as well as on instruction following (IFEval, +2.9 points). It validates that highly capable models still benefit from a dedicated mechanism for targeted error correction, further enhancing their reasoning and generation capabilities.

## 5.2 CODE GENERATION

We evaluate our method on conditional code generation using the Conala benchmark (Yin et al., 2018), where the task is to generate a Python snippet from a natural language prompt. We first train a conditional MDLM baseline on the Conala train split. The error predictor for our G-Star sampler is then trained on a disjoint hold-out split, also conditioned on a prompt. Further implementation details are provided in Appendix E.

Performance is measured by conditional perplexity under a pre-trained Qwen2.5B-Coder model (Hui et al., 2024). This metric evaluates the fluency and semantic relevance of the generated code snippet with respect to the input prompt. As shown in Table 2, our G-Star sampler outperforms both the MDLM and ReMDM baselines, achieving a lower (better) conditional perplexity. This confirms the effectiveness of our guided approach for structured generation tasks.

## 6 CONCLUSION

We introduced G-Star, a sampling method that enables efficient error correction for masked diffusion models. By using a trained error predictor to target tokens for revision, our method outperforms standard and stochastic refinement baselines like MDLM and ReMDM in computationally constrained, few-step generation regimes. We demonstrated its effectiveness and versatility across a wide range of tasks and validated its ability to enhance a state-of-the-art 7B instruction-tuned language model. The core contribution of our work is to show that targeted, intelligent refinement is a more principled and sample-efficient approach than unguided correction, paving the way for more practical and powerful discrete diffusion models.

## REPRODUCIBILITY STATEMENT

To ensure the reproducibility of our work, we provide the source code for our samplers and error predictor training in the supplementary material. All experimental details, including dataset preprocessing, model architectures, and specific hyperparameter configurations for every table and figure, are thoroughly documented in Appendix E. Furthermore, our experiments are built upon publicly

available datasets (e.g., OpenWebText, Conala) and pre-trained model checkpoints to ensure our experimental setups are accessible and verifiable by the community.

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

## APPENDIX

# A  PROOF OF CLAIM 1

In this section, we prove Claim 1: *The Variational Lower Bound (VLB) for the star-shaped process simplifies to a weighted cross-entropy objective, which has the same functional form as the standard masked diffusion objective but with different timestep-dependent weights.*

We begin with the standard VLB formulation for a non-Markovian process, which seeks to maximize the log-likelihood $\log p_\theta(\mathbf{x}_0)$:

$$\log p_\theta(\mathbf{x}_0) \geq \mathbb{E}_{q(\mathbf{x}_{1:T}|\mathbf{x}_0)} \left[ \log \frac{p_\theta(\mathbf{x}_{0:T})}{q(\mathbf{x}_{1:T} \mid \mathbf{x}_0)} \right] =: \mathcal{L}_{\text{VLB}} \tag{7}$$

By parameterizing the generative process as $p_\theta(\mathbf{x}_{0:T}) = p(\mathbf{x}_T) \prod_{t=1}^{T} p_\theta(\mathbf{x}_{t-1} \mid \mathbf{x}_t)$ and defining our star-shaped forward process as $q(\mathbf{x}_{1:T} \mid \mathbf{x}_0) = \prod_{t=1}^{T} q(\mathbf{x}_t \mid \mathbf{x}_0)$, we can decompose the VLB:

$$\mathcal{L}_{\text{VLB}} = \mathbb{E}_q \left[ \log p_\theta(\mathbf{x}_0 \mid \mathbf{x}_1) - \sum_{t=2}^{T} \text{KL} \left( q(\mathbf{x}_{t-1} \mid \mathbf{x}_0) \big\| p_\theta(\mathbf{x}_{t-1} \mid \mathbf{x}_t) \right) \right] - \text{KL}(q(\mathbf{x}_T \mid \mathbf{x}_0) \| p(\mathbf{x}_T)). \tag{8}$$

The final term is a constant with respect to the model parameters $\theta$ and can be ignored during optimization. The first term, $\mathbb{E}_q[\log p_\theta(\mathbf{x}_0 \mid \mathbf{x}_1)]$, is a reconstruction term, which already matches the form of a cross-entropy loss. Our goal is to show that the summation of KL divergence terms can also be simplified into this form.

Let's analyze a single KL divergence term from the summation for a given timestep $t$:

$$\mathcal{L}_t = \text{KL} \left( q(\mathbf{x}_{t-1} \mid \mathbf{x}_0) \big\| p_\theta(\mathbf{x}_{t-1} \mid \mathbf{x}_t) \right). \tag{9}$$

We substitute the definitions for the distributions involved:

- The true posterior is $q(\mathbf{x}_{t-1} \mid \mathbf{x}_0) = \text{Cat}(\mathbf{x}_{t-1}; \alpha_{t-1}\mathbf{x}_0 + (1 - \alpha_{t-1})\mathbf{m})$.
- The model's reverse transition is $p_\theta(\mathbf{x}_{t-1} \mid \mathbf{x}_t) = q(\mathbf{x}_{t-1} \mid \mathbf{x}_0 = \hat{\mathbf{x}}_0)$, where $\hat{\mathbf{x}}_0 = f_\theta(\mathbf{x}_t, t)$. This gives $p_\theta(\mathbf{x}_{t-1} \mid \mathbf{x}_t) = \text{Cat}(\mathbf{x}_{t-1}; \alpha_{t-1}\hat{\mathbf{x}}_0 + (1 - \alpha_{t-1})\mathbf{m})$.

The KL divergence is therefore between two categorical distributions of the same family, both of which are interpolations between a one-hot vector (the true $\mathbf{x}_0$ or the predicted $\hat{\mathbf{x}}_0$) and the mask token $\mathbf{m}$. The KL term becomes:

$$\mathcal{L}_t = \sum_{v=1}^{|V|} (\alpha_{t-1} x_{0,v} + (1 - \alpha_{t-1})m_v) \log \frac{\alpha_{t-1} x_{0,v} + (1 - \alpha_{t-1})m_v}{\alpha_{t-1} \hat{x}_{0,v} + (1 - \alpha_{t-1})m_v} \tag{10}$$

Since $\mathbf{x}_0$ is a one-hot vector corresponding to a non-mask token (let's say at index $k$), and $\mathbf{m}$ is a one-hot vector for the mask token, this sum simplifies significantly.

$$\mathcal{L}_t = \alpha_{t-1} \log \frac{\alpha_{t-1}}{\alpha_{t-1}\hat{x}_{0,k}} + (1 - \alpha_{t-1}) \log \frac{1 - \alpha_{t-1}}{1 - \alpha_{t-1}} \tag{11}$$

$$= \alpha_{t-1} \log \frac{1}{\hat{x}_{0,k}} + 0 \tag{12}$$

$$= -\alpha_{t-1} \log \hat{x}_{0,k} \tag{13}$$

Since $\hat{x}_{0,k}$ is the probability assigned by the model $f_\theta(\mathbf{x}_t, t)$ to the true token, this is precisely the negative log-likelihood, or cross-entropy loss:

$$\mathcal{L}_t = -\alpha_{t-1} \log p_\theta(\mathbf{x}_0 \mid \mathbf{x}_t). \tag{14}$$

Now, we can substitute this simplified form back into the full VLB expression from Eq. equation 8. The loss to be minimized (the negative VLB) is:

$$\mathcal{L}_{\text{final}} = -\mathcal{L}_{\text{VLB}} \approx \mathbb{E}_q \left[ -\log p_\theta(\mathbf{x}_0 \mid \mathbf{x}_1) + \sum_{t=2}^{T} \mathcal{L}_t \right] \tag{15}$$

$$= \mathbb{E}_q \left[ -\log p_\theta(\mathbf{x}_0 \mid \mathbf{x}_1) - \sum_{t=2}^{T} \alpha_{t-1} \log p_\theta(\mathbf{x}_0 \mid \mathbf{x}_t) \right] \tag{16}$$

This is a sum of weighted cross-entropy losses. By writing this as a single expectation over a timestep $t$ sampled uniformly from $\{1, \ldots, T\}$, we get:

$$\mathcal{L}_{\text{final}} = \mathbb{E}_{t \sim \mathcal{U}, \mathbf{x}_0, \mathbf{x}_t} \left[ -w'_t \log p_\theta(\mathbf{x}_0 \mid \mathbf{x}_t) \right] \tag{17}$$

where $w'_t$ are new, time-dependent weights derived from the coefficients (e.g., $w'_1 = 1$, and $w'_t = \alpha_{t-1}$ for $t > 1$, before normalization). This confirms that the training objective for the star-shaped process simplifies to the same functional form as the standard masked diffusion objective, completing the proof.

## B RELATED WORKS

### B.1 DISCRETE DIFFUSION

Discrete diffusion extends denoising ideas from continuous domains (Sohl-Dickstein et al., 2015; Ho et al., 2020; Song et al., 2020) to categorical spaces by defining token-level noising/posterior transitions. Early work formalized absorbing/structured corruption for tokens (Austin et al., 2021a; Campbell et al., 2022), while ratio-estimation and reparameterized objectives improved likelihoods and stability for text/code (Lou et al., 2024; Zheng et al., 2023; Zhao et al., 2024a). Alternative transport in discrete spaces—discrete flow matching and discrete flows—offers non-diffusive paths and guidance interfaces (Gat et al., 2024; Campbell et al., 2024; Nisonoff et al., 2024). Practical samplers and correctors further enhance decoding (Lezama et al., 2023; Zhao et al., 2024b), and analyses clarify properties of absorbing processes and conditional distributions (Ou et al., 2024).

### B.2 TEXT LATENT DIFFUSION

A complementary line runs diffusion in continuous text spaces. Diffusion-LM denoises word embeddings and enables controllable text via plug-and-play guidance (Li et al., 2022). Two-stage latent approaches compress sequences with a pretrained autoencoder and diffuse in the compact latent space, improving quality and reducing the step budget across unconditional and conditional tasks (Lovelace et al., 2024; Meshchaninov et al., 2025; Shabalin et al., 2025).

### B.3 MASKED DIFFUSION

Masked-token diffusion adopts BERT-style prediction with iterative unmasking for fast, parallel generation. MaskGIT established the paradigm on tokenized images with few refinement rounds (Chang et al., 2022). For language, simple absorbing-mask diffusion with a clean training recipe narrows the perplexity gap to autoregressive LMs and supports flexible (semi-)autoregressive decoding (Sahoo et al., 2024a). A simplified continuous-time view yields a weighted cross-entropy ELBO and state-dependent masking schedules that improve text and discrete-image modeling (Shi et al., 2024). Inference-time revision/guidance further boosts quality: remasking enables iterative correction (Wang et al.), discrete guidance improves controllability (Schiff et al., 2024; Nisonoff et al., 2024), informed correctors sharpen updates (Zhao et al., 2024b), hybrids expand self-correction regimes (von Rütte et al., 2025), and analyses examine time-agnostic behavior and sampling (Zheng et al., 2024).

### B.4 LARGE LANGUAGE DIFFUSION MODELS

Scaling diffusion for language shows competitive likelihoods and strong zero-shot behavior while keeping parallel decoding. GPT-2–scale masked diffusion narrows the gap to autoregressive transformers (Sahoo et al., 2024a), and simplified objectives with state-dependent schedules push performance and practicality (Shi et al., 2024). Recent efforts include blockwise decoding between AR and diffusion (Arriola et al., 2025), inference-time scaling/steering (Ma et al., 2025; Singhal et al., 2025), and domain-focused large models for reasoning/coding (Nie et al., 2025; Zhao et al., 2025). Prominent systems include *LLaDA* (Nie et al., 2025), *Dream 7B* (Ye et al., 2025a), and *DiffuCoder* (Gong et al., 2025).

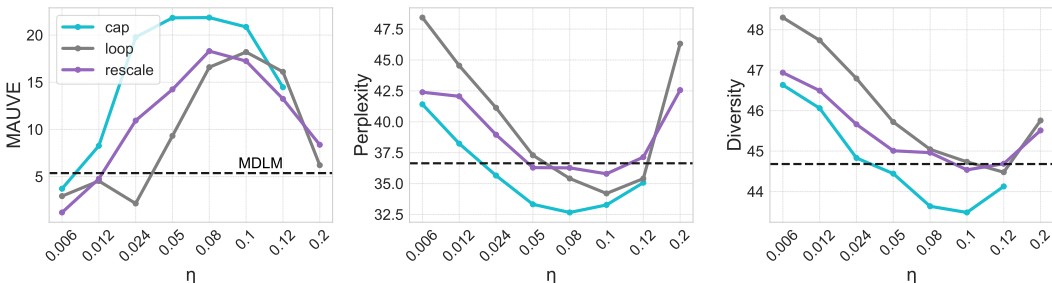

Figure 6: **Performance of ReMDM as a function of the hyperparameter $\eta$.** Results are shown for three different remasking schedules. The dashed line indicates the performance of the baseline MDLM sampler. The plots reveal a high sensitivity to the choice of $\eta$, with suboptimal values often performing worse than the baseline.

## C  ADDITIONAL RESULTS

### C.1  HYPERPARAMETER TUNING IN REMDM

A significant practical limitation of the ReMDM sampler is its high sensitivity to the remasking hyperparameter, $\eta$. This sensitivity makes the sampler's performance brittle and necessitates a costly, non-trivial tuning process to achieve any benefit over simpler baselines. In this section, we empirically quantify this dependency and demonstrate that the optimal configuration for $\eta$ is not universal, but must be determined independently for each remasking schedule.

To analyze this, we conduct an ablation study on the value of $\eta$ for three different remasking schedules proposed by the authors: 'cap', 'loop', and 'rescale'. The experiments are performed on the OWT dataset, generating sequences of length 512 with 128 sampling steps.

The results, presented in Figure 6, confirm our hypothesis and reveal two significant drawbacks of the ReMDM approach. First, performance across all metrics (MAUVE, Perplexity, and Diversity) is extremely sensitive to the choice of $\eta$. As the plots demonstrate, the relationship is non-monotonic and exhibits a "sweet spot"; a small deviation from this optimal value can cause a dramatic drop in performance. Crucially, an improper configuration of $\eta$ can render the remasking mechanism actively detrimental, with performance falling significantly below that of the standard, non-refining MDLM baseline.

Second, the optimal value for $\eta$ is not universal but must be independently and carefully tuned for each remasking schedule. Our ablation reveals that the optimal setting for the 'cap' and 'rescale' schedules is $\eta = 0.08$, whereas the 'loop' schedule achieves its peak performance at $\eta = 0.1$. This necessity for an extensive, per-schedule hyperparameter search represents a significant practical limitation, as it requires numerous runs to find a configuration that provides a tangible benefit over simpler baselines. This motivates our work on a star-shape sampler that is inherently more robust and efficient.

### C.2  ABLATION ON LOOP SIZE

As previously described, refinement process consists of: (1) an initial drafting phase with MDLM, (2) an iterative refinement "loop" at a fixed noise level, and (3) a final completion phase with MDLM. In this section, we investigate how the number of steps allocated to the refinement loop (the "loop size") affects generation quality.

**Setup.**  For this experiment, we generate 512-token sequences from OWT. The steps are allocated as follows: 115 steps for the initial MDLM draft, 13 steps for the final completion, and a variable number of steps for the central refinement loop. We vary them across a predefined grid and compare the performance of our guided sampler (G-Star-loop) against its unguided counterpart (Star-loop).

**Results.**  The results, presented in Figure 7, reveal several key dynamics of the refinement process.

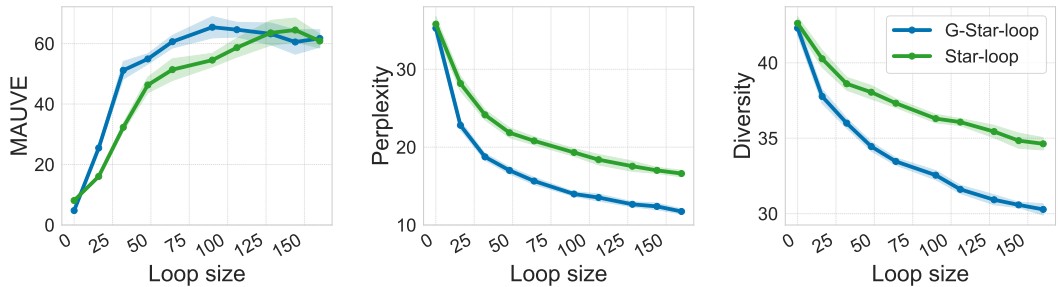

Figure 7: Performance as a function of the refinement loop size. Increasing the refinement budget generally improves quality (lower PPL, higher MAUVE) but reduces diversity. Our guided G-Star-loop demonstrates a much steeper rate of improvement, achieving higher quality with fewer steps. The MAUVE score eventually peaks and declines as the loss of diversity outweighs quality gains.

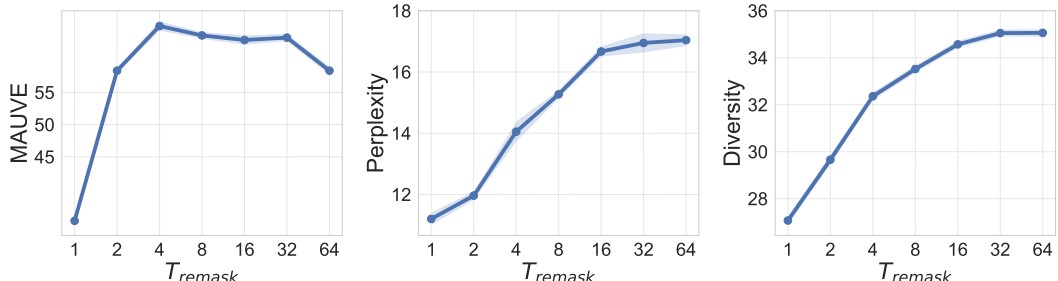

Figure 8: **Performance as a function of the error predictor temperature ($T_{\text{remask}}$).** The plots reveal a clear trade-off: lower temperatures improve quality (PPL) at the cost of diversity, while higher temperatures increase diversity at the cost of quality. The MAUVE score, which balances both, peaks at an optimal temperature of $T \approx 4 - 32$.

First, for both samplers, increasing the number of refinement steps generally leads to higher-quality text, as evidenced by a monotonic decrease in perplexity and an initial rise in the MAUVE score. This confirms the efficacy of iterative refinement. However, our guided G-Star-loop is substantially more sample-efficient, achieving a much steeper improvement curve. It consistently reaches a higher quality ceiling with fewer refinement steps compared to the unguided Star-loop.

Second, the refinement process introduces a clear trade-off between quality and diversity. As shown in the rightmost panel, a larger loop size consistently leads to a reduction in sample diversity for both methods. This can be interpreted as the model converging towards higher-quality modes in the data distribution, pruning away "noisy" or less coherent generations, but at the risk of reducing overall variety.

Finally, this quality-diversity tension directly explains the behavior of the MAUVE score. As MAUVE balances both aspects, it initially rises with perplexity improvements but eventually peaks and begins to decline as the loss in diversity becomes too significant. This phenomenon is not an artifact of a specific sampler but appears to be an inherent property of intensive iterative refinement itself: with enough steps, any refinement process will inevitably improve perplexity at the cost of diversity.

### C.3   THE QUALITY-DIVERSITY TRADE-OFF: TUNING THE ERROR PREDICTOR TEMPERATURE

We analyze the effect of the error predictor's temperature, $T_{\text{remask}}$, a hyperparameter that scales the logits from $g_\phi$ before sampling. This temperature effectively controls the stochasticity of the remasking process, acting as an intuitive control knob for the sampler's behavior. The experiment is conducted on the OWT dataset (512 tokens), using our guided sampler with the parameter-efficient error predictor configuration (a frozen 12-block MDLM backbone with a trainable classification head).

Table 3: Unconditional generation results on OpenWebText for 512-token sequences, comparing various samplers and error predictor architectures. For each column, the **best** result is marked in bold, and the second-best is underlined.

| Method | Steps = 128 | | | Steps = 256 | | | Steps = 512 | | |
|---|---|---|---|---|---|---|---|---|---|
| | MAUVE ↑ | PPL ↓ | DIV ↑ | MAUVE ↑ | PPL ↓ | DIV ↑ | MAUVE ↑ | PPL ↓ | DIV ↑ |
| MDLM | 5.4 | 36.6 | 44.7 | 2.7 | 31.1 | 40.5 | 5.1 | 27.0 | 37.8 |
| ReMDM-conf | 5.8 | 41.3 | 47.9 | 12.2 | 36.6 | 45.5 | 13.2 | 35.0 | **45.1** |
| ReMDM-cap$_{\eta=0.008}$ | 1.6 | 41.1 | 46.4 | 18.4 | 34.4 | 45.1 | 42.7 | 29.0 | 43.4 |
| ReMDM-cap$_{\eta=0.08}$ | 23.4 | 33.5 | 43.8 | 46.1 | 24.1 | 40.0 | 46.5 | 17.6 | 35.7 |
| ReMDM-loop$_{\eta=0.008}$ | 2.8 | 45.1 | **48.2** | 4.4 | 38.1 | **46.3** | 25.9 | 31.9 | 44.8 |
| ReMDM-loop$_{\eta=0.1}$ | 18.1 | 34.9 | 44.7 | 44.7 | 26.7 | 42.1 | **67.7** | 20.7 | 39.3 |
| ReMDM-rescale$_{\eta=0.015}$ | 2.0 | 64.7 | 45.1 | 10.3 | 54.3 | 43.6 | 22.7 | 45.3 | 41.7 |
| ReMDM-rescale$_{\eta=0.08}$ | 16.1 | 36.4 | 44.8 | 34.7 | 28.4 | 41.7 | 46.9 | 22.0 | 38.5 |
| Star-loop | 18.1 | 34.9 | 44.7 | 44.7 | 26.7 | 42.1 | **67.7** | 20.7 | 39.3 |
| Star+$_{t_{on}=0.2}$ | 11.7 | 32.4 | 41.5 | 34.3 | 24.5 | 38.3 | 45.5 | 14.7 | 31.3 |
| G-Star-loop$_{1B,F}$ | 44.8 | 19.7 | 35.3 | **65.0** | 14.7 | 31.4 | 56.3 | 11.6 | 27.5 |
| G-Star-loop$_{12B,H}$ | **57.3** | 18.4 | 36.2 | 63.8 | 14.1 | 32.4 | 50.1 | 10.7 | 27.9 |
| G-Star-loop$_{12B,F}$ | **57.3** | **17.2** | 35.4 | 60.9 | **12.7** | 30.9 | 58.6 | **9.9** | 26.4 |
| G-Star+$_{t_{on}=0.2,12B}$ | 40.1 | 19.5 | 36.5 | 51.6 | 14.4 | 31.7 | 48.9 | 10.7 | 26.9 |

The results, presented in Figure 8, reveal a clear and non-monotonic relationship between the predictor temperature and the overall generation quality as measured by MAUVE. This behavior is a direct consequence of an underlying trade-off between sample quality (Perplexity) and Diversity, which the temperature directly controls.

At low temperatures (e.g., $T = 1$), the predictor's output becomes more deterministic, focusing the remasking on a small set of tokens with the highest predicted error probability. This leads to highly precise corrections of the most obvious errors, resulting in the best perplexity scores (middle panel). However, this precision comes at the cost of significantly reduced diversity (right panel), as the sampler explores a much narrower set of possible revisions.

Conversely, as the temperature increases, the error probabilities become more uniform. In this regime, the guided sampler's behavior converges towards that of the unguided, Star-loop sampler. This predictably increases sample diversity but degrades perplexity, as the error correction is no longer targeted and becomes less effective.

The MAUVE score, which balances both quality and diversity, peaks at a temperature of $T \approx 4-32$. At this point, the sampler achieves an optimal balance between precise error correction and sufficient generative variety. This analysis highlights that the predictor temperature serves as an important lever for controlling the generation process, allowing practitioners to tailor the sampler's behavior: lower temperatures can be used for high-fidelity tasks where correctness is paramount, while higher temperatures may be preferable for creative tasks where diversity is the primary goal.

### C.4 UNCONDITIONAL GENERATION RESULTS ON OPENWEBTEXT

Complementary to Figure 5 in the main text, Table 3 summarizes extended unconditional generation results on OpenWebText for 512-token sequences. We adopt the full suite of ReMDM (Wang et al.) sampling protocols and report each both with the authors' original remasking hyperparameters and with tuned alternatives, to provide a faithful comparison: ReMDM-cap with $\eta = 0.008$ (original) and $\eta = 0.08$ (tuned), ReMDM-loop with $\eta = 0.008$ (original) and $\eta = 0.1$ (tuned), and ReMDM-rescale with $\eta = 0.015$ (original) and $\eta = 0.08$ (tuned), alongside ReMDM-conf as introduced in prior work; a broader discussion on hyperparameter sensitivity in ReMDM can be found in Appendix C.1.

Complementary to Figure 4, we additionally include Star+ and G-Star+ variants with $t_{on} = 0.2$ to contrast our optimal loop-like scheduler with simpler cap-like refinement strategy under the same computational budgets.

Table 4: Results for an error predictor trained on OWT and evaluated on out-of-domain datasets. For each evaluation dataset, we report the validation perplexity (PPL val) of the OWT-pretrained diffusion model and the binary classification metrics of the error predictor.

| Evaluation dataset | Domain | PPL val | Accuracy | AUC-ROC |
|---|---|---|---|---|
| OWT | General | $\leqslant 22.89$ | 0.88 | 0.94 |
| TinyStories | Stories | $\leqslant 12.72$ | 0.88 | 0.98 |
| OpenWebMath | Math | $\leqslant 33.72$ | 0.85 | 0.92 |
| CNN/DailyMail | News | $\leqslant 25.69$ | 0.88 | 0.94 |
| The Stack | Python code | $\leqslant 31.96$ | 0.84 | 0.90 |

### C.5 ROBUSTNESS OF THE ERROR PREDICTOR

In the experiments above, the error predictor was both trained and evaluated on the same dataset. An important question, however, is how sensitive the predictor is to the choice of training data and to what extent it generalizes to unseen domains. To investigate this, we trained the error predictor on the OWT dataset and evaluated it on several datasets spanning diverse domains:

- TinyStories Eldan & Li (2023): a synthetic dataset of short stories written in simple English at the comprehension level of a typical 3–4-year-old child.

- OpenWebMath Paster et al. (2023): a dataset of high-quality mathematical text filtered from Common Crawl.

- CNN/DailyMail Hermann et al. (2015): a dataset of news articles authored by professional journalists at CNN and the Daily Mail.

- The Stack Kocetkov et al. (2022): a large-scale code dataset containing over 6 TB of source code in 358 programming languages, from which we use only the Python subset.

To assess both the diffusion model's performance and the error predictor's generalization, we randomly sampled 10,000 examples from each evaluation dataset. We analyze the diffusion model's behavior on these new domains by reporting its validation perplexity (PPL val). We then evaluate the OWT-trained error predictor's ability to identify these errors using two standard binary classification metrics: Accuracy (at a 0.5 probability threshold) and AUC-ROC. The AUC-ROC score is particularly relevant as it measures the predictor's quality across all thresholds, which is crucial given that our G-Star sampler does not rely on a fixed threshold.

The results, presented in Table 4, show several key trends. The TinyStories dataset appears to be the simplest case for both the diffusion model and the error predictor, which is evident from its minimal validation perplexity (12.72) and the predictor's near-perfect AUC-ROC score (0.98). Conversely, the CNN/DailyMail dataset seems closest to the OWT training data, as its validation perplexity (25.69) is near the OWT baseline (22.89), and it achieves an identical AUC-ROC score (0.94). Unsurprisingly, the most challenging domains for both models are the specialized OpenWebMath and The Stack (Python code) datasets, which show higher perplexity and slightly lower predictor performance. Overall, however, the drop in predictor quality across these diverse domains is not severe. This suggests that the error predictor successfully learns general patterns of diffusion errors, allowing it to generalize effectively even when trained on only a single dataset.

## D COMPUTATIONAL OVERHEAD

**Time overhead.** We quantify the deployment cost of the different samplers by measuring end-to-end wall-clock latency on OpenWebText with sequence length $L = 512$ and $T \in \{128, 256, 512\}$ diffusion steps. All runs use batch size 1 on a single H200 GPU. Table 5 reports, for each method, the total generation time per 512-token sample and the corresponding number of effective Transformer forward passes (NFEs), where one NFE denotes a single pass of the full backbone over a length-$L$ sequence.

Table 5: Wall-clock generation time and number of Transformer forward passes (NFEs) per 512-token sample on OpenWebText (batch size 1). Times are reported as mean $\pm$ standard deviation in seconds.

| Method | Steps $T$ | Time [s] | NFEs |
|---|---|---|---|
| AR (KV cache) | 512 | $2.28 \pm 0.03$ | 512 |
| AR (no KV cache) | 512 | $2.51 \pm 0.01$ | 512 |
| MDLM / ReMDM / Star | 128 | $2.26 \pm 0.09$ | 128 |
| G-Star-loop$_{1B,F}$ | 128 | $2.60 \pm 0.10$ | 133 |
| G-Star-loop$_{12B,H}$ | 128 | $3.43 \pm 0.10$ | 192 |
| G-Star-loop$_{12B,F}$ | 128 | $3.43 \pm 0.10$ | 192 |
| G-Star+$_{t_{\text{on}}=0.2,12B}$ | 128 | $2.71 \pm 0.10$ | 154 |
| MDLM / ReMDM / Star | 256 | $4.56 \pm 0.03$ | 256 |
| G-Star-loop$_{1B,F}$ | 256 | $5.24 \pm 0.02$ | 267 |
| G-Star-loop$_{12B,H}$ | 256 | $6.96 \pm 0.07$ | 384 |
| G-Star-loop$_{12B,F}$ | 256 | $6.96 \pm 0.07$ | 384 |
| G-Star+$_{t_{\text{on}}=0.2,12B}$ | 256 | $5.49 \pm 0.06$ | 308 |
| MDLM / ReMDM / Star | 512 | $9.16 \pm 0.07$ | 512 |
| G-Star-loop$_{1B,F}$ | 512 | $10.44 \pm 0.08$ | 533 |
| G-Star-loop$_{12B,H}$ | 512 | $13.93 \pm 0.08$ | 768 |
| G-Star-loop$_{12B,F}$ | 512 | $13.93 \pm 0.08$ | 768 |
| G-Star+$_{t_{\text{on}}=0.2,12B}$ | 512 | $10.84 \pm 0.11$ | 615 |

As autoregressive (AR) baselines we use a GPT-2 small (Radford et al., 2019) whose parameter counts are matched to MDLM backbone. We consider a latency-optimized AR model with key–value (KV) caching, and a variant that recomputes the full prefix at every step ("AR-w/o KV"). The latter is closer to our diffusion setting, where each update operates on the full sequence, and helps disentangle the effect of cache reuse from the intrinsic cost of masked diffusion. In principle, similar KV-based accelerations (e.g., (Wu et al., 2025)) could also be adapted for masked diffusion; see Appendix H for a broader discussion of diffusion speed-up techniques.

For the diffusion baselines, MDLM, ReMDM, and Star all use the same backbone with no auxiliary networks. Their compute therefore coincides, with $\text{NFE}_{\text{MDLM}} = \text{NFE}_{\text{ReMDM}} = \text{NFE}_{\text{Star}} = T$; Star only changes the masking policy and introduces no extra passes. G-Star augments this baseline with guidance that is active only on a subset of the trajectory. Let $\Delta = t_{\text{off}} - t_{\text{on}}$ be the fraction of guided steps, and let $D$ denote the number of Transformer blocks in the backbone while $B$ is the number of blocks used by the predictor. Measured in units of a full-depth pass, each guided step then contributes $(B/D)$ additional NFEs, so the total cost is

$$\text{NFE}_{\text{G-Star}} = T + T\Delta\frac{B}{D} = \left(1 + \Delta\frac{B}{D}\right)T.$$

In our configurations, the backbone has $D = 12$ blocks. G-Star-loop$_{12B}$ uses a full-depth predictor ($B = 12$) with $t_{\text{on}} = 0.55, t_{\text{off}} = 0.05$, giving NFE $= 1.5\,T$; G-Star-loop$_{1B}$ uses a single-block predictor ($B = 1$) with the same $t_{\text{on}} = 0.55, t_{\text{off}} = 0.05$, yielding NFE $= \left(1 + 0.5 \cdot \frac{1}{12}\right)T \approx 1.04\,T$; and G-Star$^+$ uses a full-depth predictor ($B = 12$) with $t_{\text{on}} = 0.2, t_{\text{off}} = 0$, giving NFE $= 1.2\,T$. The measured wall-clock times in Table 5 closely follow these ratios: the G-Star variants incur a controlled $(1 + \Delta B/D)$-factor overhead.

**Memory overhead.** Star uses exactly the same backbone and parameterization as the underlying MDLM and therefore has no memory overhead. For G-Star, peak activation memory is essentially unchanged: the predictor operates on the current logits and is run sequentially after the base diffusion step, so we do not need to keep additional large activations in memory, only per-token error scores and the index set of remasked positions.

The remaining overhead comes from parameters. In the G-Star-loop$_{12B,F}$ variant we store an additional full 12-block Transformer as the predictor. In the G-Star-loop$_{1B,F}$ variant we only add a single Transformer block. In the parameter-efficient G-Star-loop$_{12B,H}$ variant we do not store a second backbone at all and only add a small token-wise classification head on top of the existing model.

Table 6: Benchmark-specific sequence lengths and the noise level $\alpha_{on}$ for the refinement loop.

| Benchmark | Sequence Length ($L$) | $\alpha_{on}$ |
|---|---|---|
| MMLU | 128 | 0.88 |
| MMLU-PRO | 128 | 0.88 |
| GSM8K | 256 | 0.95 |
| GPQA | 128 | 0.88 |
| HumanEval | 768 | 0.98 |
| MBPP | 1024 | 0.98 |
| IFEval | 1280 | 0.98 |

# E IMPLEMENTATION DETAILS

## E.1 UNCONDITIONAL TEXT GENERATION ON OPENWEBTEXT

For this experiment, we closely follow the original MDLM setup for the backbone, and only add a linear head for error predictor training.

**Error predictor head training.** The error predictor head, $g_\phi$, was trained for 50,000 steps on a single H200 GPU approximately 48 hours using a global batch size of 512. For optimization, we used AdamW with a learning rate of 1e-4. The learning rate was managed by a constant schedule with warmup with 2,500 warmup steps.

## E.2 CODE GENERATION ON CONALA

**Dataset and preprocessing.** We use the Conala benchmark (Yin et al., 2018), which contains Python code snippets paired with natural language intents. We construct our dataset splits as follows: the train set consists of 2,000 curated samples plus 594,000 samples from the mined subset; the hold-out set for training the error predictor contains 380 curated samples; and the test set contains 500 samples. All prompts and code snippets were tokenized using the `gpt-2` tokenizer, with sequences truncated or padded to a maximum length of 128 tokens.

**MDLM baseline training.** Our baseline is a conditional Masked Diffusion Language Model (MDLM), following the 12-layer Transformer architecture of Sahoo et al. (2024a). We employed a two-stage training procedure. First, the model was pre-trained for 50,000 steps on the full train set (mined and curated combined) with a batch size of 1024. Subsequently, the model was fine-tuned for an additional 10,000 steps exclusively on the curated portion of the training data, using a smaller batch size of 512. For both stages, we used the AdamW optimizer with a learning rate of 3e-4.

**Error predictor training.** The error predictor, $g_\phi$, for our G-Star sampler was trained on the hold-out split. We employed a parameter-efficient setup: the predictor's backbone consists of the full 12-layer transformer from our trained conditional MDLM with its weights frozen. We then added a single linear classification head on top of the final layer's token representations, and trained only this head to predict token-level errors, conditioned on the same prompts. We used the AdamW optimizer with a learning rate of 3e-4 and a batch size of 380. The model was trained for 500 steps. The training takes 1 hour on a single H200 GPU.

## E.3 LARGE-SCALE EXPERIMENTS

This section details the experimental setup used for the large-scale evaluation on the Dream-Instruct 7B model, with results presented in Table 1.

**Models and benchmarks.** We use the publicly available Dream-Instruct 7B model as our base model. The evaluation is conducted on a diverse suite of seven benchmarks: MMLU (Hendrycks et al., 2020), MMLU-PRO, GSM8K (Cobbe et al., 2021), GPQA (Rein et al., 2023), HumanEval (Chen et al., 2021), MBPP (Austin et al., 2021b), and IFEval. The sequence length for each benchmark is specified in Table 6.

**Baseline sampling configuration.** Our reproduced baseline follows the official configuration from the Dream repository. The sampling process consists of a number of steps equal to the sequence length ($T = $ seq_len), where one mask is denoised at each step. The diffusion temperature is set to 0.1. The confidence score for selecting which token to unmask is calculated as the entropy of the predicted logits, as specified by their 'alg=entropy' setting.

**G-Star sampler configuration.** For our method, we augment the baseline setup by integrating a loop-based refinement strategy within the same computational budget. The total number of sampling steps is kept identical to the baseline ($T = $ seq_len), but we repurpose 10% of these steps for refinement. Specifically, 90% of the steps are used for standard progressive denoising, while the remaining 10% are dedicated to refinement loops where, at each step, we remask $N = 15$ tokens identified as errors by our predictor. All other parameters, such as the diffusion temperature (0.1) and the base confidence metric (entropy), are kept identical to the baseline for a fair comparison. The error predictor temperature was set to 0.

**Error predictor training.** The error predictor $g_\phi$ for our G-Star sampler was trained on the Tulu 3 dataset (Lambert et al., 2024). We employed a parameter-efficient strategy: the predictor's backbone consists of the full, frozen Dream-Instruct 7B model. We added a lightweight classification head, consisting of an RMSNorm layer and a linear layer, and trained only this head. The predictor was trained for 70k steps on 8 H200 GPUs over 24 hours. We used a global batch size of 128 and the AdamW optimizer with a learning rate of $3 \times 10^{-4}$, $\beta_1 = 0.9$, $\beta_2 = 0.999$, and a constant learning-rate schedule with a warmup of 5000 steps.

## F    VISUALIZING THE REFINEMENT PROCESS

To provide a qualitative and intuitive understanding of the difference between our guided sampler and its unguided counterpart, we visualize their remasking behavior over the course of a full generation. The following figures illustrate the set of masked tokens (orange dots) at each step of the generation process for a 512-token sequence generated in 256 steps. Both processes are divided into two distinct phases: an initial MDLM drafting phase (steps 0-113), where tokens are progressively unmasked, a subsequent refinement phase (steps 114-240), where remasking occurs, and final MDLM generation phase (steps 241-256).

**Analysis of remasking strategies.** A direct comparison of Figure 10 and Figure 9 reveals the fundamental difference between the two refinement strategies. The unguided sampler operates via a stochastic, unstructured process, treating all tokens as equally likely candidates for revision. In stark contrast, our guided sampler demonstrates an intelligent and structured approach. The error predictor identifies and clusters likely errors, enabling the sampler to perform more meaningful revisions. The emergence of "continuous segments" in the guided plot is particularly significant; it provides strong qualitative evidence that our method moves beyond simple token-level fixes and is capable of performing coherent, phrase-level refinement, a feat that is extremely improbable under the indiscriminate selection of the unguided approach.

**Text generation example of G-Star sampler.** In addition, in this section, we provide examples of text generation using G-Star and the unguided Star. Figure 11 provides a visual snapshot of the refinement process, showing steps 90 through 95. Both G-Star+ (top) and the unguided Star+ (bottom) begin this phase with an identical text draft. The tokens highlighted in red are those selected for remasking at each step.

A clear difference in strategy is immediately visible. The unguided Star+ sampler (bottom) exhibits an unfocused, token-level remasking, selecting apparently random tokens for revision (e.g., steps 91 and 94). This indiscriminate approach is inefficient, as it may remask already correct tokens while failing to target problematic phrases. In stark contrast, G-Star+ (top) demonstrates a more structured and intelligent approach. Guided by the error predictor, it identifies and remasks semantically problematic regions. For example, in the transition from step 90 to 91, it targets the weak phrase "a period and mostly silent" for a coherent, phrase-level revision, resulting in "that we are". This tar-

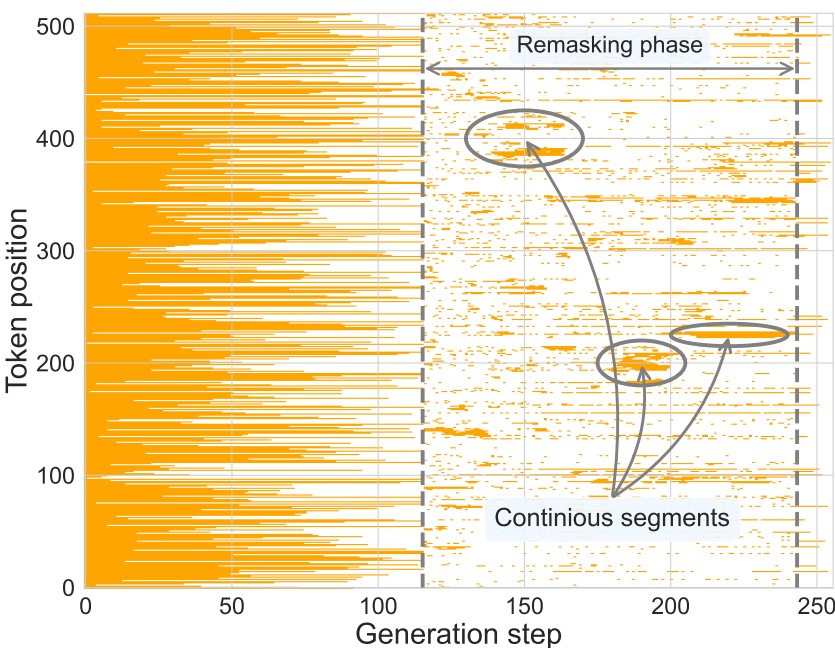

Figure 9: Remasking pattern of the guided G-Star-loop sampler. The plot visualizes the masked token positions (orange dots) at each generation step. In contrast to the unguided sampler, our guided approach exhibits a highly structured remasking pattern. The error predictor directs the sampler to focus on specific, clustered regions of the text. This often results in the selection of **contiguous segments** for revision, as highlighted in the figure. This ability to perform coordinated, phrase-level corrections is a key advantage of our targeted approach.

geted correction is highly unlikely to occur with the random sampling of Star+ and allows G-Star+ to perform more efficient, surgical edits to improve text quality.

## G  PRACTICAL GUIDANCE FOR HYPERPARAMETER TUNING

This section provides practical guidance for readers who wish to apply our method and select appropriate hyperparameters. The two most important parameters are the remasking schedule (i.e., when to apply G-Star) and the sampling temperatures.

**Remasking schedule.**  As we demonstrate in Section 4.2, text generation via masked diffusion can be broadly divided into two phases: an initial **context accumulation phase** and a subsequent **text refinement phase**. This two-stage structure aligns with practical observations from previous work on remasking, such as ReMDM (Wang et al.). Based on this insight, we recommend enabling the remasking process (i.e., refinement) only toward the end of the generation, once the model has already formed a coherent draft. In our experiments, we typically activated the remasking schedule within the noise level range of $t \in [0.1, 0.3]$, for both the G-Star-loop and G-Star+ strategies. For optimal results on a specific task, we recommend tuning this starting timestep $t$ as a key hyperparameter.

**Sampling temperatures.**  The second set of critical hyperparameters involves the diffusion and predictor temperatures. As discussed in Section 4.4, increasing the **diffusion model's temperature** produces more diverse and varied token predictions. This can be strategically advantageous: one can use a higher diffusion temperature to encourage *exploration* (proposing a wider set of token options), while relying on the error predictor to *filter* these proposals and retain only the correct ones. Separately, the **error predictor's temperature** (analyzed in Section C.3) controls its confidence. Lowering the predictor's temperature makes it more conservative, i.e., it will only remask tokens

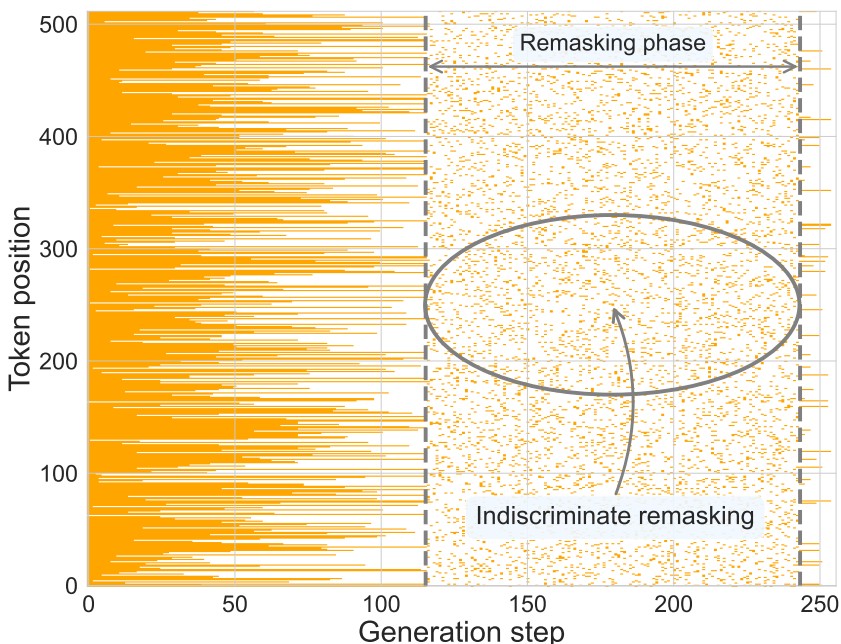

Figure 10: Remasking pattern of the Unguided Star-loop sampler. The plot visualizes the masked token positions (orange dots) at each generation step. During the remasking phase (steps 114-240), the pattern of selected tokens is scattered and visually resembles random noise. This illustrates the indiscriminate nature of the unguided approach, where every token position has a roughly equal probability of being revised at each step, without regard to the semantic or syntactic structure of the text.

that it is highly confident are incorrect. However, since the predictor is not perfect and can also make mistakes, setting this temperature to an extremely low value (e.g., zero) may not be ideal. We recommend using a non-zero temperature to balance the predictor's precision and recall.

## H    LIMITATIONS

Despite its effectiveness, our method has three primary limitations. First, our framework is restricted to "in-place" token substitution and cannot perform insertion or deletion operations. This means that while the model can correct a token by changing its value (e.g., 'house' → 'home'), it cannot correct an error of omission by inserting a new token between two existing ones, as this would require shifting the entire subsequent sequence. Extending the framework to predict and apply "shift" or "insert/delete" operations is a promising direction for future work.

Second, the error predictor requires a separate, sequential training stage, which adds complexity to the overall training pipeline. This could potentially be addressed by exploring methods for jointly training the main diffusion model and the error predictor in an end-to-end fashion, which might also foster a tighter synergy between the generation and refinement processes.

Finally, our study does not attempt to aggressively optimize inference throughput. All samplers are evaluated using a straightforward implementation that does not exploit recent acceleration techniques for diffusion LLMs, such as KV-caching (Wu et al., 2025), self-speculative decoding (Gao et al., 2025), or enhanced forms of parallel decoding like adaptive parallel decoding (Israel et al., 2025). Integrating G-Star with these complementary methods represents an exciting direction for future work and could further narrow the remaining gap to highly optimized autoregressive systems.

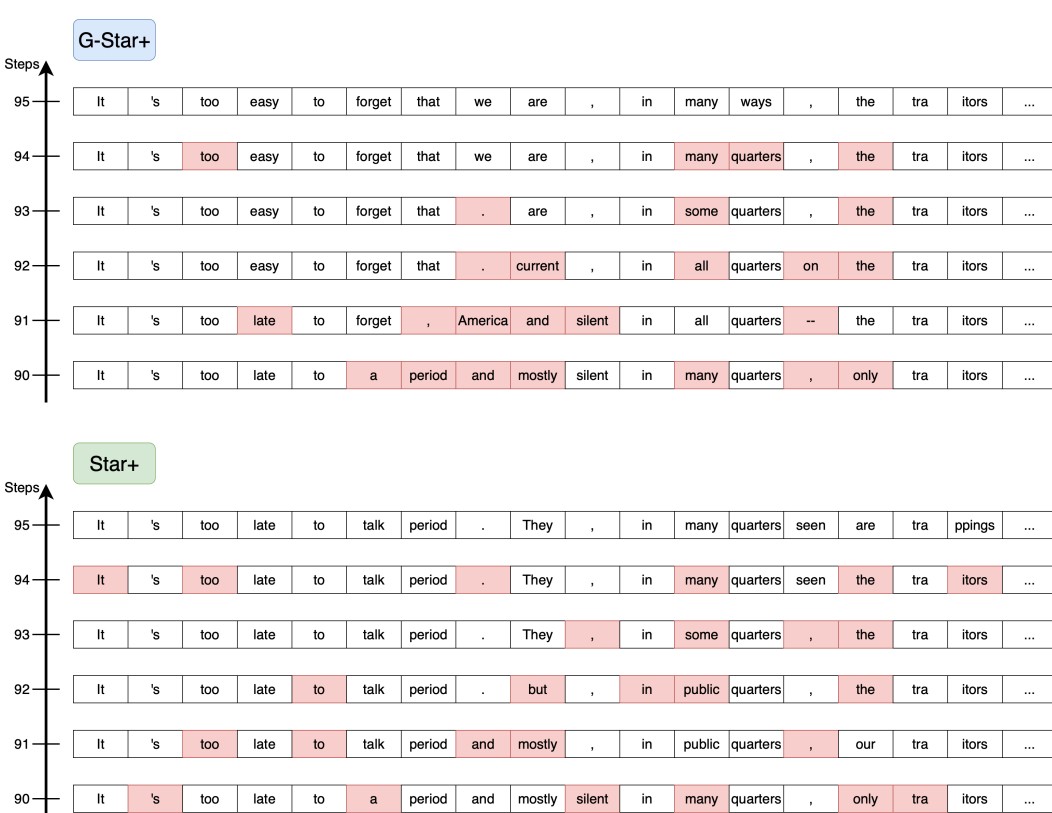

Figure 11: The figure provides a snapshot of the refinement process, showing steps 90 through 95 from a 128-step generation of a 512-token sequence. Both the unguided Star+ (bottom) and our guided G-Star+ (top) begin this phase with an identical text draft generated by a standard MDLM. The panels display the beginning of the text sequence, with tokens remasked at each step highlighted in red. Starting from the same draft, the two methods immediately diverge. The Star+ sampler exhibits an indiscriminate, token-level remasking strategy that appears unfocused. In contrast, our G-Star+ sampler demonstrates a more structured approach.

