# OpenReview forum: "Guided Star-Shaped Masked Diffusion"
_ICLR.cc/2026/Conference — Submitted to ICLR 2026_

### Official Review · Reviewer_pM5q · 2025-10-27

**Soundness:** 3
**Presentation:** 3
**Contribution:** 3
**Rating:** 6
**Confidence:** 3

**Summary:**

The paper proposes Guided Star-Shaped Masked Diffusion (G-Star), a new sampling framework for masked diffusion language models that enables iterative error correction. The method has two components: (1) a "star-shaped" formulation where all latent states $q(x_t|x_0)$ connect directly to the clean data $x_0$ rather than forming a Markov chain, allowing non-monotonic transitions and token revision; (2) a lightweight learned error predictor $g_φ$ that identifies likely incorrect tokens for targeted remasking. The authors prove the training objective remains a weighted cross-entropy (Claim 1), ensuring compatibility with pre-trained MDLMs. Experiments on OpenWebText unconditional generation and seven downstream benchmarks with Dream-Instruct-7B show consistent improvements, especially in few-step regimes (32-256 steps).

**Strengths:**

1. The star-shaped formulation is theoretically elegant and practically compatible with existing models. Claim 1 proves that despite the non-Markovian forward process in Equation 4, the VLB simplifies to weighted cross-entropy identical to MDLM training, enabling direct reuse of pre-trained weights. The formulation also subsumes ReMDM as a special case.
2. The error predictor design is parameter-efficient and practical. Table 1 shows a 1-block predictor nearly matches the 12-block version, and head-only training matches full fine-tuning. For Dream-Instruct-7B, freezing the 7B backbone and training only a linear head make large-scale deployment feasible with minimal training overhead.
3. The empirical results are strong across diverse settings. In few-step regimes, G-Star+ achieves MAUVE >40 at 128 steps versus ~10 for Star+ (Figure 3), and G-Star-loop reaches 57.3 MAUVE at 128 steps versus 23.4 for best ReMDM (Table 1). Benefits extend to Dream-Instruct-7B with gains on all seven benchmarks including MMLU +1.3, GPQA +1.8, and IFEval +2.9 (Table 3), plus improved code generation with 17.8 perplexity at 64 steps versus 22.5 for ReMDM on Conala (Table 2).
4. The experimental validation is comprehensive with systematic ablations. Section 4.2 identifies optimal t_on ≈ 0.3 and explains why pure samplers fail (Figure 2). Section 4.3 shows guided sampling consistently outperforms unguided remasking with largest gaps in the 64-256 step regime (Figure 3). Section 4.5 confirms lightweight predictors suffice, establishing both effectiveness and practical efficiency.

**Weaknesses:**

1. The paper lacks wall-clock comparisons to autoregressive baselines. While G-Star outperforms diffusion models, it requires T sampling steps versus one AR pass. The paper only measures efficiency in step count, not actual wall-clock time for deployment.
2. The early-phase instability requires a hybrid schedule. The method does not stand on its own over the full trajectory: pure star-shaped sampling degrades text and needs MDLM drafting before refinement.
3. This method has imited edit expressivity. The sampler only does in-place substitution; it cannot insert or delete tokens. That constrains the kind of structural edits it can correct, which may matter for code or long-form generation.
4. The error predictor is trained by simulating denoising and labeling token errors against ground truth (Algorithm 1). Each experiment uses task-specific training (OWT for OWT, Tulu 3 for Dream-Instruct), but the paper never evaluates cross-domain robustness or addresses how the ground-truth error labeling handles instruction-following tasks with multiple valid outputs

**Questions:**

N/A

---

> ### Author Response · Authors · 2025-11-23
> **Appreciation and new pareto front analysis**
>
> We sincerely thank you for the careful and positive assessment of our submission. We are glad you found the motivation, the star-shaped formulation, and the lightweight error predictor compelling, and we appreciate your summary of the empirical gains.
>
> In the time since our initial submission, we conducted additional experiments to more comprehensively characterize the trade-offs inherent in generative refinement. We recognized that evaluating at fixed hyperparameters provides only a partial view. To address this, we performed a new analysis plotting the Pareto frontiers of perplexity versus diversity (by varying the diffusion model's temperature). These new plots, which we have added to the paper (Section 4.4, Figure 5), offer a much clearer picture of the performance landscape. We hope you might find a moment to review these new results, as they significantly strengthen our paper's contributions.
>
> These Pareto fronts (Figure 5) clearly demonstrate the superiority of our guided refinement. G-Star-loop consistently dominates the unguided ReMDM-loop sampler and the MDLM baseline, establishing a more favorable trade-off between perplexity and diversity at every computational budget. Strikingly, G-Star-loop in the highly constrained 128-step regime manages to achieve both lower perplexity and greater diversity than the best-performing ReMDM-loop sampler running for 512 steps. This result strongly validates our hypothesis that guided, targeted refinement is fundamentally more efficient than unguided stochastic remasking.
>
> We now turn to address each of your specific concerns.

---

> > ### Author Response · Authors · 2025-11-23
> > **W1. On analysis of wall-clock latency**
> >
> > Thank you for emphasizing wall-clock latency as the practical metric for deployment. We fully agree that reporting only the number of diffusion steps can be incomplete. To address this directly, we ran end-to-end timing benchmarks for all methods under identical settings and added a dedicated latency analysis in Appendix D. For convenience, we reproduce the core timing table for generating a single sample on OpenWebText (sequence length L=512, batch size 1). Following standard practice, we compare to autoregressive decoding with KV-cache, since this is the realistic deployment baseline.
> >
> > | Method | Steps $T$ | Time [s] | NFEs |
> > | --- | --- | --- | --- |
> > | AR | 512 | 2.28 $\pm$ 0.03 | 512 |
> > | MDLM / ReMDM / Star | 128 | 2.26 $\pm$ 0.09 | 128 |
> > | G-Star-loop$_{1B, F}$ | 128 | 2.60 $\pm$ 0.10 | 133 |
> > | G-Star-loop$_{12B, H}$ | 128 | 3.43 $\pm$ 0.10 | 192 |
> > | G-Star-loop$_{12B, F}$ | 128 | 3.43 $\pm$ 0.10 | 192 |
> > | G-Star$_{ton =0.2,12B}$ | 128 | 2.71 $\pm$ 0.10 | 154 |
> > | MDLM / ReMDM / Star | 256 | 4.56 $\pm$ 0.03 | 256 |
> > | G-Star-loop$_{1B, F}$ | 256 | 5.24 $\pm$ 0.02 | 267 |
> > | G-Star-loop$_{12B, H}$ | 256 | 6.96 $\pm$ 0.07 | 384 |
> > | G-Star-loop$_{12B, F}$ | 256 | 6.96 $\pm$ 0.07 | 384 |
> > | G-Star$_{ton=0.2,12B}$ | 256 | 5.49 $\pm$ 0.06 | 308 |
> > | MDLM / ReMDM / Star | 512 | 9.16 $\pm$ 0.07 | 512 |
> > | G-Star-loop$_{1B, F}$ | 512 | 10.44 $\pm$ 0.08 | 533 |
> > | G-Star-loop$_{12B, H}$ | 512 | 13.83 $\pm$ 0.08 | 768 |
> > | G-Star-loop$_{12B, F}$ | 512 | 13.83 $\pm$ 0.08 | 768 |
> > | G-Star$_{ton=0.2,12B}$ | 512 | 10.84 $\pm$ 0.08 | 615 |
> >
> > First, we observe that despite achieving the same or better quality, the 128-step G-Star samplers require **2–3× less wall-clock time** than the corresponding 512-step ReMDM runs. This highlights the efficiency benefit of our targeted guided refinement: strong sample quality can be reached using a significantly smaller computational budget.
> >
> > Second, in comparison to autoregressive decoding, it is important to note that AR models rely on KV-cache by default, whereas our current implementation does not use any caching mechanism. Even so, the 128-step G-Star variants operate in a similar runtime range. At the same time, there is still meaningful room to speed up our approach: recent works have demonstrated how diffusion LMs can benefit from KV-cache–style optimizations [1, 2]. These techniques improve the global sampler efficiency, and are fully complementary to our contribution, which focuses on token-wise refinement. Exploring these combinations is a very promising direction for future work.

---

> > > ### Author Response · Authors · 2025-11-23
> > > **W2. On the schedule hyperparameters**
> > >
> > > We agree that the need to choose a hybrid schedule (when to switch from MDLM to the star-shaped sampler and how many refinement steps to allocate) is a limitation. However, we view this as a small practical cost relative to the gains we obtain without training a completely new model from scratch.
> > >
> > > At the same time, this hybrid setup is what makes our method flexible: it allows us to take **any pre-trained masked diffusion LM** and immediately use it for guided refinement **without additional finetuning**, which is a major practical advantage. Switching from drafting to refinement also does not require any extra computation beyond choosing this one hyperparameter. To simplify its use, we added a recommendation section (Appendix G) that explains the intuition behind the default hyperparameters and how to adjust them in practice.

---

> > > > ### Author Response · Authors · 2025-11-23
> > > > **W3. On the limited edit expressivity**
> > > >
> > > > We acknowledge that the current sampler cannot dynamically change sequence length and only performs in-place substitutions, which limits expressivity. However, even within this constraint, the method can still perform non-local edits, as the error predictor is able to select contiguous spans for remasking, effectively rewriting clauses rather than individual tokens (see Figures 9 and 11 in the Appendix F). We agree that supporting true variable-length editing through explicit insertion and deletion operations is indeed an important and promising direction for future work, and we explicitly highlight this in the Limitations section.

---

> > > > > ### Author Response · Authors · 2025-11-23
> > > > > **W4. On the cross-domain robustness**
> > > > >
> > > > > We sincerely thank the reviewer for the very helpful suggestion to directly test the cross-domain robustness of our error-predictor. As was also noted by reviewer ocUj, our original ablation setup lacked a dedicated robustness analysis. To address this, we conducted a new experiment (Appendix C.5) aimed specifically at evaluating how well the predictor generalizes beyond its training distribution. We took the same diffusion model and error-predictor trained on OpenWebText (context length 512) used in the main paper and evaluated it on several unseen datasets: TinyStories, OpenWebMath, CNN/DailyMail, and The Stack. For the reviewer’s convenience, we replicate the corresponding robustness table below.
> > > > >
> > > > > | Evaluation dataset | Domain | PPL val | Accuracy | AUC-ROC |
> > > > > | --- | --- | --- | --- | --- |
> > > > > | OWT | General | $\leqslant 22.89$ | 0.88 | 0.94 |
> > > > > | TinyStories | Stories | $\leqslant 12.72$ | 0.88 | 0.98 |
> > > > > | OpenWebMath | Math | $\leqslant 33.72$ | 0.85 | 0.92 |
> > > > > | CNN/DailyMail | News | $\leqslant 25.69$ | 0.88 | 0.94 |
> > > > > | The Stack | Python code | $\leqslant 31.96$ | 0.84 | 0.90 |
> > > > >
> > > > > Across all out-of-domain datasets, the predictor maintains strong accuracy and AUC-ROC despite substantial variation in style, difficulty, and domain. The performance on TinyStories and CNN/DailyMail remains close to the in-domain baseline, while the drop on more challenging datasets such as OpenWebMath and The Stack is moderate and follows the expected increase in backbone perplexity. Overall, these findings indicate that the error-predictor learns generalizable patterns of diffusion errors rather than memorizing dataset-specific artifacts. We include this new results and a broader discussion in the revised version of the manuscript.

---

> > > > > > ### Author Response · Authors · 2025-11-23
> > > > > > **References**
> > > > > >
> > > > > > [1] Wu et al. "Fast-dllm: Training-free acceleration of diffusion llm by enabling kv cache and parallel decoding.” (2025).
> > > > > >
> > > > > > [2] Wu et al. "Fast-dLLM v2: Efficient Block-Diffusion LLM." (2025).

---

> > > > > > > ### Author Response · Authors · 2025-11-23
> > > > > > >
> > > > > > > Thank you for your valuable feedback and suggestions. If our responses have addressed your concerns, we would kindly appreciate your consideration of revising your score. Please do not hesitate to let us know if any points require further clarification.

---

### Official Review · Reviewer_ocUj · 2025-10-29

**Soundness:** 3
**Presentation:** 3
**Contribution:** 2
**Rating:** 6
**Confidence:** 3

**Summary:**

The paper proposes Guided Star-Shaped Masked Diffusion (G-Star), which aims to address the core limitations of discrete masked diffusion models, namely the irreversibility of decisions during sampling and the lack of error correction capability. The authors point out that although existing approaches (e.g., ReMDM) introduce a re-masking mechanism, they rely on random strategies, resulting in low efficiency. In contrast, G-Star employs a star-shaped sampling structure coupled with a learnable error predictor to achieve targeted and efficient error correction, leading to significantly improved generation quality, particularly in few-step sampling scenarios.

**Strengths:**

1. The paper clearly identifies a fundamental limitation of discrete masked diffusion—irreversible token commitments—and provides a well-motivated, elegant solution.
2. The proposed star-shaped formulation is theoretically sound and compatible with existing pretrained MDLMs, avoiding retraining.
3. The guided remasking mechanism (error predictor) is lightweight yet effective, enabling targeted error correction and strong empirical improvements.
4. Extensive experiments (text, code, large models) and clear ablation studies convincingly demonstrate robustness and practical value.

**Weaknesses:**

1. The paper is conceptually dense; the star-shaped reparameterization could be explained more intuitively for accessibility.
2. Relying solely on pseudocode and textual descriptions may not adequately convey the paper. It is recommended that the authors provide an integrated methodological figure to help readers quickly grasp the proposed approach.
3. The method’s sensitivity to hyperparameters is discussed but might benefit from deeper theoretical justification or generalization guidelines.

**Questions:**

1. How robust is the learned error predictor when applied to unseen domains or tasks with different token distributions?
2. Could the proposed guidance mechanism be integrated into autoregressive diffusion hybrids for further speed–quality trade-offs?

---

> ### Author Response · Authors · 2025-11-23
> **Thank you for the positive feedback**
>
> Thank you for your review and positive assessment of our work. We greatly appreciate that you recognized the motivation behind our design choices and the value of our ablation studies. We address your comments below.

---

> > ### Author Response · Authors · 2025-11-23
> > **W1, W2. On a more intuitive method presentation**
> >
> > Thank you for these valuable suggestions. We made a deliberate effort to present the method clearly, but we agree that the star-shaped sampling can benefit from a more intuitive exposition and a dedicated figure.
> >
> > In the original submission, we included an illustrative figure clarifying the overall procedure in the appendix. Following your recommendation, we have moved this figure into the main paper (now Figure 1). We have also revised the accompanying text (Section 3) to use this figure to intuitively walk the reader through the star-shaped and guided star-shaped sampling procedures in a high-level way, before delving into the pseudocode.
> >
> > We hope these changes directly address your concerns and make the method's presentation much clearer and more accessible. We are, of course, very open and grateful for any further suggestions if you feel the clarity could be improved even more.

---

> > > ### Author Response · Authors · 2025-11-23
> > > **W3. On hyperparameter guidelines**
> > >
> > > We agree that guidance on hyperparameters is crucial for reproducibility and practical use. To address this, we have added a new dedicated section, "Practical guidance for hyperparameter tuning" (Appendix G), to help practitioners and future users of our algorithm find the optimal settings for their specific applications.
> > >
> > > This new section offers concrete recommendations, supported by the intuition and empirical justification behind the key hyperparameters. Specifically, we explain:
> > >
> > > 1.  When to apply remasking, linking this decision to the distinct context accumulation (early) and text refinement (late) phases of generation.
> > > 2.  How to use the two main temperatures (for the denoiser versus the predictor) to practically manage the quality-versus-diversity trade-off.
> > >
> > > We believe this guidance makes our algorithm more accessible and easier for users to apply effectively.

---

> > > > ### Author Response · Authors · 2025-11-23
> > > > **Q1. On the robustness of the error predictor**
> > > >
> > > > We agree that evaluating the predictor's robustness is crucial, a point also raised by Reviewer pM5q. We tested the error predictor, which was trained on the OWT dataset, on four out-of-distribution datasets with different token distributions: TinyStories [1] (children’s stories), OpenWebMath [2] (mathematical text), CNN/DailyMail [3] (news), and The Stack [4] (Python code).
> > > >
> > > > To assess both the diffusion model's performance and the error predictor's generalization, we randomly sampled 10,000 examples from each evaluation dataset. We analyze the diffusion model's behavior on these new domains by reporting its validation perplexity (PPL val). We then evaluate the OWT-trained error predictor's ability to identify these errors using two standard binary classification metrics: Accuracy (at a 0.5 probability threshold) and AUC-ROC. The AUC-ROC score is particularly relevant as it measures the predictor's quality across all thresholds, which is crucial given that our G-Star sampler does not rely on a fixed threshold.
> > > >
> > > > We have added a new section to the appendix (Appendix C.5) with a full analysis. For your convenience, we reproduce the main results here:
> > > >
> > > > | Evaluation dataset | Domain | PPL val | Accuracy | AUC-ROC |
> > > > | --- | --- | --- | --- | --- |
> > > > | OWT | General | $\leqslant 22.89$ | 0.88 | 0.94 |
> > > > | TinyStories | Stories | $\leqslant 12.72$ | 0.88 | 0.98 |
> > > > | OpenWebMath | Math | $\leqslant 33.72$ | 0.85 | 0.92 |
> > > > | CNN/DailyMail | News | $\leqslant 25.69$ | 0.88 | 0.94 |
> > > > | The Stack | Python code | $\leqslant 31.96$ | 0.84 | 0.90 |
> > > >
> > > > The results show several key trends. The TinyStories dataset appears to be the simplest case for both the diffusion model and the error predictor, which is evident from its minimal validation perplexity ($12.72$) and the predictor's near-perfect AUC-ROC score ($0.98$). Conversely, the CNN/DailyMail dataset seems closest to the OWT training data, as its validation perplexity ($25.69$) is near the OWT baseline ($22.89$), and it achieves an identical AUC-ROC score ($0.94$). Unsurprisingly, the most challenging domains for both models are the specialized OpenWebMath and The Stack (Python code) datasets, which show higher perplexity and slightly lower predictor performance. Overall, however, the drop in predictor quality across these diverse domains is not severe. This suggests that the error predictor successfully learns general patterns of diffusion errors, allowing it to generalize effectively even when trained on only a single dataset.

---

> > > > > ### Author Response · Authors · 2025-11-23
> > > > > **Q2. On integration with autoregressive–diffusion hybrids**
> > > > >
> > > > > Architectures that combine autoregressive and diffusion-style generation span a broad design space, and we are not entirely sure which specific class of “hybrids” you have in mind. If you are referring to diffusion models with semi-autoregressive or block-wise sampling, then our guidance mechanism is in principle compatible: within each block we can apply guided diffusion sampling independently. A careful exploration of this integration, however, would require substantial additional experimentation and is therefore beyond the scope of the current paper. If you had a different family of hybrids in mind, we would be grateful for further clarification and would be pleased to provide a more detailed answer.

---

> > > > > > ### Author Response · Authors · 2025-11-23
> > > > > > **References**
> > > > > >
> > > > > > [1] Eldan et al. "Tinystories: How small can language models be and still speak coherent english?" (2023).
> > > > > >
> > > > > > [2] Paster et al. "Openwebmath: An open dataset of high-quality mathematical web text." (2023).
> > > > > >
> > > > > > [3] Hermann et al. "Teaching machines to read and comprehend." (2015).
> > > > > >
> > > > > > [4] Kocetkov et al. "The stack: 3 tb of permissively licensed source code." (2022).

---

### Official Review · Reviewer_jgo2 · 2025-10-31

**Soundness:** 3
**Presentation:** 3
**Contribution:** 3
**Rating:** 6
**Confidence:** 3

**Summary:**

This paper addresses the irreversible sampling procedure in masked diffusion models, which constrains performance in low-step generation regimes. It introduces G-Star, a novel sampling algorithm that reformulates generation using a star-shaped paradigm to allow for error correction and augments it with a learnable re-masking scheduler to intelligently revise likely errors.

**Strengths:**

- The paper identifies a fundamental limitation of masked diffusion models: the sequence of irreversible commitments imposes a ceiling on sample quality.
- The proposed enables iterative refinement with a learned scheduler for targeted error correction.
- The method demonstrates a commanding advantage in low-step regimes, underscoring the critical importance of intelligent guidance when refinement opportunities are limited.

**Weaknesses:**

- The conceptual building blocks (star-shaped process, guidance, loop) are adapted from prior work; the primary novelty lies in their specific application to error correction in masked discrete diffusion.
- The error predictor $g_{\phi}$ may face a potential distribution shift, as it is trained on single-step denoised data but applied to iteratively refined, model-generated drafts during inference.
- The optimal switch-over time ($t_{on}$) was determined for the unguided sampler but reused for the guided sampler without specific validation for the latter.
- There are minor clarity and presentation issues: $g_{\phi}$ is used before being formally defined, and tables are placed out of order (Table 3 is referenced before Table 2).
- Several key analyses, parameter settings, and overhead claims are not fully substantiated in the main text. For instance, the claim of "negligible increase in inference overhead" lacks a quantitative analysis of wall-clock time or memory, and important details like the sensitivity analysis for $T_{remask}$ (Appendix C.3) and loop schedule specifics (Appendix C.2) are confined to the appendices, hindering the main text's self-containedness and the readability.

**Questions:**

- Could the authors comment on the potential distribution shift for the error predictor between training and inference, and its impact on performance during iterative refinement?
- Can the authors provide a concrete analysis of the inference overhead (wall-clock time, peak memory) compared to the baseline MDLM and unguided Star+ samplers?
- Have the authors validated that the optimal $t_{on}$ found for the unguided Star+ sampler is also optimal for the guided G-Star+ sampler, or could performance be further improved by tuning it for the guided setting?

---

> ### Author Response · Authors · 2025-11-23
> **Appreciation and new pareto front analysis**
>
> We sincerely thank you for your review and thoughtful assessment of our work. We are particularly grateful for your recognition of our method's core contribution and for highlighting its ``commanding advantage in low-step regimes.''
>
> Since submitting this work, we conducted further analysis that shows our method's advantages are even more general. Rather than relying on point-wise comparisons at fixed hyperparameters, which only provide a limited view, we performed a more thorough evaluation by plotting the Pareto fronts for perplexity versus diversity (by varying the diffusion model's temperature). These new results are now included in the paper (Section 4.4, Figure 5).
>
> The Pareto fronts in Figure 5 highlight the clear advantage of our guided refinement strategy. Across all computational budgets, G-Star-loop consistently achieves a superior balance between perplexity and diversity, outperforming both the MDLM baseline and the unguided ReMDM-loop remasking sampler. Remarkably, even in the severely constrained 128-step regime, G-Star-loop attains lower perplexity while simultaneously delivering higher diversity than the best-tuned ReMDM configuration with 512 steps. These findings confirm that targeted refinement, rather than stochastic remasking, is essential for efficient quality improvements.
>
> This new analysis confirms that while our guided error predictor is highly effective in low-step regimes (as you correctly noted), it also achieves the best overall trade-off between quality and diversity even at higher computational budgets.

---

> ### Author Response · Authors · 2025-11-23
> **W1. On novelty of star-shaped process, guidance, and loop.**
>
> We would like to clarify the relationship between our work and the star-shaped process introduced by Ohotin et al. [1]. Ohotin et al. proposed to consider a generative process in which the transition to $x_{t-1}$ depends only on the model's prediction
> $\hat{x}_0$ and is conditionally independent of $x_t$, in other words:
>
> $$p_{\theta} (x_{t-1} \mid x_t) = q(x_{t-1} \mid \hat{x}_0).$$
>
> In their case, making the process work required incorporating sufficient statistics of the tail distribution, as well as complete retraining of the model, and also the use of a completely different noising schedule, as described in Section 3 of their work.
>
> While our work is inspired by the general intuition behind the star-shaped process, we address the challenge of applying this sampler in a substantially different way. We show that applying such a process across the full trajectory is ineffective, and we provide empirical justification for using it only toward the end of the trajectory. Futhermore, we theoretically show that our method can be combined with pretrained masked diffusion models, such as MDLM, without any additional finetuning. This is not possible in the framework of Ohotin et al. [1].
>
> Additionally, to the best of our knowledge, the idea of a learnable re-masking mechanism has not previously been explored in either continuous or discrete diffusion models, and we consider this a key contribution of our work.
>
> Regarding the loop schedule, we fully agree with the reviewer’s observation. We follow the schedule used in ReMDM, as it has been shown to be highly effective; we explicitly note this in Section 4.4. At the same time, our experiments indicate that guided re-masking provides substantial improvements even when using only a quarter of the diffusion steps (Figure 5).

---

> ### Author Response · Authors · 2025-11-23
> **W2, Q1. On distribution shift between training and inference**
>
> Thank you for raising this important and widely discussed question. The training–inference discrepancy has received significant attention in the broader generative modeling literature. In autoregressive models, this is known as exposure bias [2], where models are trained with ground-truth conditioning but generate from their own predictions at inference. Similar concerns arise in diffusion models [3, 4] and even discrete diffusion models [5].
>
> However, while several techniques have been proposed to address this mismatch, these approaches are not commonly adopted in practice: the improvements they offer are often modest, while the computational overhead is substantial. Therefore, in this work, we did not focus on this effect. Instead, our primary investigation was whether guided remasking improves text generation and to determine the optimal conditions for its use.
>
> While our work did not focus on this effect, we agree it is an important question. In response to your feedback, we have now conducted an additional experiment to investigate this mismatch, following the approach proposed for discrete diffusion models in [5]. The experimental setup is as follows. For each clean sample $x_0$, we:
>
> 1. apply the standard noising process to obtain $x_t$,
> 2. denoise once to get $\hat{x}_0$,
> 3. re-noise $\hat{x}_0$ using the **error predictor** to obtain $\hat{x}_t$,
> 4. denoise again to obtain $\hat{\hat{x}}_0$,
> 5. train the error predictor predict $\hat{\hat{x}}_0 \neq x_0$.
>
> This reduces the gap between training distributions and inference distributions by ensuring the model regularly encounters “model-generated” imperfect drafts.
>
> ### **Results**
>
> Below is the comparison between the standard training setup and the reduced train–inference mismatch setup:
>
> | Model                     | MAUVE $\uparrow$  | PPL $\downarrow$ | DIV $\uparrow$ |
> |---------------------------|-----------------|-----------------|--------------|
> | G-Star-loop, 128          | 57.3            | 17.2            | **35.4**         |
> | + reduced mismatch        | **59.7**        | **17.0**       | 34.90    |
> | G-Star-loop, 256          |  60.9           |  12.7          | **30.9**          |
> | + reduced mismatch        |  **61.7**       |  12.7          | 30.7    |
> | G-Star-loop, 512          |  58.6           |  9.9           | **26.4**          |
> | + reduced mismatch        |  **60.6**       |**9.8**         | 26.2    |
>
> ### **Interpretation**
>
> The performance differences are relatively small. While there are slight changes in some metrics, the improvements are not significant. At the same time, training becomes **roughly twice as slow**, as each step requires additional denoising and remasking operations.
>
> This mirrors findings in previous work: mitigating the mismatch is theoretically appealing, but the practical gains are often limited relative to the added computational cost.
>
> Finally, we note that diffusion models in general are trained on real data and applied to self-generated samples at inference. Despite this intrinsic mismatch, they remain effective and robust in practice. Our results suggest the same holds in our setting.
>
> Nevertheless, we agree that this is a meaningful and challenging direction. A more systematic exploration of robust remasking under controlled distribution shift is an excellent topic for future work, and we plan to investigate it further.

---

> ### Author Response · Authors · 2025-11-23
> **W3, Q3. On the choice of the optimal switch-over time**
>
> Thank you for this precise and important question regarding hyperparameter tuning.
>
> You are correct that we did not perform a separate, exhaustive search for the optimal $t_{sw}$ specifically for G-Star. Our primary goal was not to find the absolute best hyperparameter for every individual experiment, but rather to demonstrate the general effectiveness of the guided remasking approach compared to the unguided one, even when using a $t_{sw}$ optimized for the baseline.
>
> As we demonstrate in Section 4.2, the refinement process is most effective when activated within a general time range (e.g., $t \in [0.1, 0.4]$). The parameter $t_{sw}$ we used falls within this effective range and proved to be robust, yielding strong results for G-Star across all tasks presented in the paper.
>
> We agree that, as is common practice, this hyperparameter would likely benefit from being tuned specifically for any new task to ensure optimal results. To that end, we have now added a new section in the Appendix (see Appendix G) with practical recommendations and intuition for hyperparameter selection. This new section is intended to help practitioners and future users of our algorithm find the optimal settings for their specific applications.

---

> ### Author Response · Authors · 2025-11-23
> **W4. On typos**
>
> Thank you for pointing out these presentation issues. We have corrected these in the revised manuscript.

---

> ### Author Response · Authors · 2025-11-23
> **W5. On information placed in the appendix**
>
> Thank you for this comment. Due to the strict page limit, we had to move secondary ablations and implementation details to the appendix. We prioritized the core contributions for the main paper:
> 1.  Explaining the Star/G-Star methodology.
> 2.  Analyzing how diffusion behaves across different trajectory segments (a phenomenon not previously shown for text diffusion).
> 3.  Demonstrating that our method works with pretrained masked diffusion models without additional finetuning.
>
> The appendix contains supporting material such as the sensitivity analysis for $T_{\text{remask}}$, loop-schedule details, and extended ablations. We agree that some readers might prefer to see more of this in the main text; however, due to the space constraints, we were forced to prioritize. We do provide clear references in the main paper to these important appendix sections for readers who wish to delve into the details.
>
> Furthermore, as mentioned in our response to W2, we have now added a new section in the appendix (Appendix G) with practical recommendations for hyperparameter tuning, to make it as straightforward as possible for others to implement and use our algorithm.

---

> ### Author Response · Authors · 2025-11-23
> **Q2. On analysis of the inference overhead**
>
> Thank you for this suggestion. We have now added a detailed analysis of the computational overhead to the paper (see Appendix D). The analysis includes wall-clock generation times, the number of transformer forward passes (NFEs), and memory overhead for each variant. This provides a comprehensive view of the practical costs associated with our method and allows for direct comparison with baseline approaches. For your convenience, we reproduce the key results here:
>
> | Method | Steps $T$ | Time [s] | NFes |
> | --- | --- | --- | --- |
> | MDLM / ReMDM / Star | 128 | 2.26 $\pm$ 0.09 | 128 |
> | G-Star-loop$_{1B, F}$ | 128 | 2.60 $\pm$ 0.10 | 133 |
> | G-Star-loop$_{12B, H}$ | 128 | 3.43 $\pm$ 0.10 | 192 |
> | G-Star-loop$_{12B, F}$ | 128 | 3.43 $\pm$ 0.10 | 192 |
> | G-Star$_{t_\mathrm{on}=0.2,12B}$ | 128 | 2.71 $\pm$ 0.10 | 154 |
> | MDLM / ReMDM / Star | 256 | 4.56 $\pm$ 0.03 | 256 |
> | G-Star-loop$_{1B, F}$ | 256 | 5.24 $\pm$ 0.02 | 267 |
> | G-Star-loop$_{12B, H}$ | 256 | 6.96 $\pm$ 0.07 | 384 |
> | G-Star-loop$_{12B, F}$ | 256 | 6.96 $\pm$ 0.07 | 384 |
> | G-Star$_{t_\mathrm{on}=0.2,12B}$ | 256 | 5.49 $\pm$ 0.06 | 308 |
> | MDLM / ReMDM / Star | 512 | 9.16 $\pm$ 0.07 | 512 |
> | G-Star-loop$_{1B, F}$ | 512 | 10.44 $\pm$ 0.08 | 533 |
> | G-Star-loop$_{12B, H}$ | 512 | 13.83 $\pm$ 0.08 | 768 |
> | G-Star-loop$_{12B, F}$ | 512 | 13.83 $\pm$ 0.08 | 768 |
> | G-Star$_{t_\mathrm{on}=0.2,12B}$ | 512 | 10.84 $\pm$ 0.08 | 615 |
>
> **1. Time overhead**
>
> * Star / ReMDM: These methods have zero time overhead compared to the MDLM baseline. They perform the exact same number of computations ($T$ forward passes); they just change the masking policy.
> * G-Star: Our most powerful G-Star-loop (using a full predictor for 50% of the steps) has a ~50% overhead.
>
> **2. Memory overhead**
>
> * Star / ReMDM: These methods have zero memory overhead.
> * G-Star:
>     * **Activation memory:** Peak activation memory is almost unchanged. This is because the error predictor runs sequentially after the main model, so we don't need to store two sets of large activations in memory at the same time.
>     * **Parameter memory:** The only overhead is in parameter memory (for storing the predictor). The model we used in our experiments on large-scale and code is a parameter-efficient version that shares the backbone and only adds a tiny classification head, resulting in almost no parameter overhead at all.
>
> In summary, G-Star offers a flexible trade-off between a controllable inference cost and significant quality gains. As our Pareto-front analysis (Figure 5) shows, G-Star for 128 steps achieves a better quality-diversity balance than unguided ReMDM with 512 steps while being **nearly 3 times faster** than the 512-step ReMDM baseline (3.43s vs. 9.16s).

---

> ### Author Response · Authors · 2025-11-23
> **References**
>
> [1] Okhotin et al. "Star-shaped denoising diffusion probabilistic models." (2023).
>
> [2] Bengio et al.  "Scheduled Sampling for Sequence Prediction with Recurrent Neural Networks." (2015).
>
> [3] Tan et al. "E2ED²: Direct Mapping from Noise to Data for Enhanced Diffusion Models." (2024).
>
> [4] Huang et al. "Self Forcing: Bridging the Train–Test Gap in Autoregressive Video Diffusion." (2025).
>
> [5] Asada et al. "Addressing the Training‑Inference Discrepancy in Discrete Diffusion for Text Generation." (2025).

---

> ### Author Response · Authors · 2025-11-23
>
> We thank you for your thoughtful feedback and suggestions. We would be grateful if you would consider raising your score, in case we have addressed your concerns. Please let us know if any aspects still need clarification.

---

### Official Review · Reviewer_sA7a · 2025-11-02

**Soundness:** 3
**Presentation:** 3
**Contribution:** 3
**Rating:** 4
**Confidence:** 3

**Summary:**

The paper proposes a novel sampling algorithm for error correction in discrete diffusion. The authors reparameterize the diffusion sampling process into a star-shaped formulation where each latent directly conditions on the clean data. They then augment this with a learned error predictor that identifies likely incorrect tokens and selectively re-masks them, enabling more efficient and powerful refinement during generation.

**Strengths:**

## Strengths
1. **Motivation & importance.** The problem is important, and the motivation for enabling targeted error correction in discrete diffusion is compelling.
2. **Clear analysis.** The analysis is intuitive and fairly comprehensive. In §4.2, the paper distinguishes **two generation phases** that emphasize different objectives and favor different sampling strategies, which is quite interesting.1. **Novelty positioning.** The core contribution needs clearer separation from prior work. **ReMDM** already explores re-masking predicted tokens for error correction; it seems the main novelty here lies in the **guided prediction module** rather than the overall idea of remasking.
2. **Generalization beyond text.** It’s unclear how the approach transfers to **vision**. Testing on an image benchmark (e.g., ImageNet) under the **ReMDM** setup would help establish applicability to the image domain.
3. **Methodological details.** More specifics are needed about training the **error predictor**—its **architecture**, **training configuration**, and **compute budget**.

3. **Downstream evaluation.** The method is evaluated on downstream tasks, including **code generation**, demonstrating practical benefits.

**Weaknesses:**

1. **Novelty positioning.** The core contribution needs clearer separation from prior work. **ReMDM** already explores re-masking predicted tokens for error correction; it seems the main novelty here lies in the **guided predictor**.  Can you clarify this?
2. **Generalization beyond text.** It’s unclear how the approach transfers to vision. Testing on an image benchmark (e.g., ImageNet) under the ReMDM setup would help establish applicability to the image domain.
3. **Methodological details.** Please include more details about training the **error predictor**, such as architecture, training configuration, and compute budget.

**Questions:**

1. **Compute overhead.** The approach adds a guided sampler to predict which tokens to re-mask. What is the **computational cost** of this component? Do you perform an additional forward pass at **every diffusion step**, and how does this scale with the number of steps and sequence length?
2. **Metric trade-offs (Table 1).** Your method excels on **MAUVE** and **PPL** at 128/256/512 steps, whereas **ReMDM** does better on **DIV**.  Is there any way to preserve the DIV?
3. **Sensitivity to fine-tuning (Table 1).** Performance appears sensitive to fine-tuning strategy. Many best results occur under different settings. Can you explain this?
4. You argue that when the sampling budget is small, each step is critical and performance degrades without the guided error predictor. However, in Table 1 the best result at 128 steps is achieved by $Star+_{ton}=0.2$ is the best, can you exlain this?

---

> ### Author Response · Authors · 2025-11-23
> **Thank you for the feedback**
>
> We sincerely thank you for the thoughtful and encouraging feedback. We are particularly pleased that you found the motivation for targeted error correction in discrete diffusion compelling, and we appreciate your positive assessment of our downstream evaluations. Below, we respond to each of your comments in detail.

---

> ### Author Response · Authors · 2025-11-23
> **W1. Novelty positioning**
>
> As you correctly point out, ReMDM [1] already explores **unguided** re-masking of predicted tokens for error correction, and we explicitly acknowledge this in the Introduction and Preliminaries. However, because the re-masking is unguided, ReMDM frequently re-masks already correct tokens, which makes refinement process inefficient and requires many sampling steps to achieve good quality.
>
> Our work addresses this in two stages:
>
> 1. **A simpler "Star-shaped" formulation (Star+, Sec. 3.1):** Before introducing any guidance, we first propose the Star-shaped formulation for generation. This formulation simplifies the re-masking process by working directly with the full prediction $\hat{\mathbf{x}}_0$ and, most importantly, it removes the complex $\sigma_t$ re-masking schedule required by ReMDM. This simplification is a contribution in itself. As we show in Appendix C.1 (Figure 6), an inappropriate choice of ReMDM's $\sigma_t$ parameters can lead to poor behavior, even compared to the MDLM baseline. Our Star+ formulation eliminates the need for this costly tuning and, as shown in Table 3 and Figure 6, its built-in choice is near-optimal, achieving performance comparable to a best-tuned ReMDM while simplifying the sampler.
>
> 2. **The guided error predictor ($g_{\phi}$), (G-Star, Sec. 3.2):** On top of this simpler Star+ framework, we then introduce our main contribution: a **learned guided predictor ($g_{\phi}$)**. This predictor is designed to estimate which tokens in $\hat{\mathbf{x}}_0$ are likely incorrect and selectively re-masks only those. This guided component directly addresses the core inefficiency of ReMDM reducing unnecessary modifications of correct tokens and making each sampling step significantly more effective.
>
> In summary, while ReMDM indeed introduced the idea of re-masking for error correction, our novelty is:
> (i) a **simpler Star+ formulation** that removes the complex $\sigma_t$ hyperparameter, and
> (ii) a **guided error predictor ($g_{\phi}$)**, built upon this simpler framework, that makes re-masking targeted and efficient, overcoming the primary limitations of unguided re-masking.

---

> ### Author Response · Authors · 2025-11-23
> **W2. On generalization beyond text**
>
> Thank you for this excellent suggestion. Per your request, we have conducted a new experiment to test our method's applicability to the image domain.
>
> We follow the ReMDM and MaskGiT [2] setup for class-conditional generation on ImageNet [3] (256x256). In this experiment, we work within the VQ-GAN [4] latent space, treating the resulting image tokens as a 1D sequence. Building on this text-like representation, we then train a lightweight linear head over MaskGiT backbone as an error predictor, a strategy consistent with our other experiments. We test our G-Star+ sampler (using a $t_{on}=0.2$ schedule) against these baselines. The results are as follows:
>
> | **Sampler** | **T = 16** | **T = 32** |
> | --- | --- | --- |
> | FID $\uparrow$ |  |  |
> | MaskGiT | 6.74 | **4.92** |
> | MDLM | 7.88 | 5.37 |
> | ReMDM$_{\eta=0.05}$ | 7.40 | **4.92** |
> | G-Star+$_{t_{on}=0.2}$ | **6.39** | 5.29 |
> | IS $\downarrow$ |  |  |
> | MaskGiT | 155.32 | 181.57 |
> | MDLM | 140.97 | 169.79 |
> | ReMDM$_{\eta=0.05}$ | 145.27 | 182.05 |
> | G-Star+$_{t_{on}=0.2}$ | **179.7** | **206.3** |
>
> Although re-masking slightly improves the results for T=16, the overall effect is less pronounced than what we observed in text generation. We hypothesize there are two primary reasons why re-masking, in general, has a less pronounced effect in this VQ-GAN setting compared to text:
>
> 1.  **Token Interchangeability:** Unlike text, the discrete latent space for images contains high redundancy. Many VQ tokens are visually similar and effectively interchangeable. This makes it difficult for a predictor to isolate a single "correct" token, as multiple tokens may be equally valid.
> 2.  **Decoder Robustness:** The VQ-GAN decoder is highly robust. Even if the diffusion model predicts a slightly "incorrect" token, the decoder can often map it to the correct visual feature. Swapping one latent token may not result in any perceptible change to the final image, making the re-masking process less impactful.
>
> In summary, while our method shows a clear benefit in the low-step regime, its overall impact is more complex in the image domain. We believe these domain-specific properties mean that a direct application of text-based re-masking is not optimal, and further dedicated research would be needed to effectively adapt this approach for VQ-based image generation.

---

> ### Author Response · Authors · 2025-11-23
> **W3. Implementations details**
>
> Thank you for this suggestion. We have updated the Implementation Details section (Appendix E) to provide a more comprehensive description of the error predictor's training.

---

> ### Author Response · Authors · 2025-11-23
> **Q2. Metric trade-offs**
>
> Thank you for this insightful question regarding the metric trade-offs. You've correctly identified that in the point-wise comparisons ReMDM appears to achieve higher diversity in some settings.
>
> We also noted this, and it motivated us to investigate this specific quality-diversity balance more thoroughly, as we felt that comparisons at fixed hyperparameters provide a limited view.
>
> To address this properly, we performed a more comprehensive evaluation, which is now included in the revised paper (Section 4.4, Figure 5). Instead of focusing on single-point metrics, we analyzed the full **Pareto front** for the perplexity-diversity trade-off by varying the diffusion model's temperature.
>
> This new analysis in Figure 5 clearly demonstrates the superiority of our guided approach. The G-Star-loop method consistently occupies a better Pareto frontier than both the MDLM baseline and the unguided ReMDM-loop, showcasing a much more favorable balance between perplexity and diversity across all compute budgets.
>
> A striking example of this is that our 128-step G-Star-loop model not only achieves lower perplexity but also simultaneously delivers higher diversity than the best-tuned 512-step ReMDM. This finding strongly suggests that targeted, guided refinement is fundamentally more efficient and effective than the stochastic remasking used by ReMDM.

---

> ### Author Response · Authors · 2025-11-23
> **Q3. Sensitivity to training strategy**
>
> Thank you for this observation. Similar to our investigation of the quality-diversity trade-off, we conducted a more thorough comparison of these fine-tuning schemes to clarify this (Section 4.5 of revised version).
>
> Specifically, we found that:
> 1.  A parameter-efficient strategy (training only a linear head on a full, frozen backbone) performs almost as well as fine-tuning the entire model.
> 2.  The major performance sensitivity you noted comes from using a **1-block predictor**. This lightweight architecture is simply not powerful enough to learn the error patterns effectively. As seen in our Pareto analysis (Figure 5b), this 1-block variant performs significantly worse than its deeper analogs and is only marginally better than the unguided baseline.
>
> Therefore, the sensitivity is less about the fine-tuning method and more about ensuring the predictor has sufficient capacity for the task.

---

> ### Author Response · Authors · 2025-11-23
> **Q4. On typographical error**
>
> We greatly appreciate your comment about Table 1 (now Table 4). You are absolutely right to point out this issue: unfortunately, we made a typographical error and accidentally swapped the MAUVE and generative perplexity values for Star+$_{ton=0.2}$ (steps=$128$) in that row. We have corrected it in the revised version of the paper.
>
> We also note that Figure 3 already shows the results for the same Star+$_{ton=0.2}$ configuration with the correct values, which we believe makes clear that the mistake was unintentional.

---

> ### Author Response · Authors · 2025-11-23
> **Q1. Compute overhead**
>
> Thank you for this question. We have now added a detailed analysis of the computational overhead to the paper (see Appendix D). The analysis includes wall-clock generation times, the number of transformer forward passes (NFEs), and memory overhead for each variant. This provides a comprehensive view of the practical costs associated with our method and allows for direct comparison with baseline approaches.
>
> To answer your question directly: the cost is controllable and is not necessarily an additional full forward pass at every step.
>
> In our proof-of-concept experiments (Section 4), we did this to show the maximum potential of guidance. In that setup, we perform an additional forward pass of the backbone on 50% of the diffusion steps to predict errors. This results in 1.5x the number of model calls compared to a baseline with the same number of generation steps.
>
> However, as we show in Figure 5, this 1.5x overhead is extremely efficient. Even with this overhead, our guided sampler using **4x fewer total diffusion steps** (e.g., 128 steps) still achieves a better quality-diversity balance than unguided ReMDM with 512 steps while being **nearly 3 times faster** than the 512-step ReMDM baseline (3.43s vs. 9.16s)
>
> Furthermore, in our large-scale experiments, we made the comparison even more direct. We used a parameter-efficient predictor where the number of main backbone calls was exactly the same as the baseline, ensuring a fair comparison.

---

> ### Author Response · Authors · 2025-11-23
> **References**
>
> [1] Wang et al. "Remasking Discrete Diffusion Models with Inference-Time Scaling." (2025).
> [2] Chang et al. "MaskGIT: Masked Generative Image Transformer." (2022).
> [3] Deng et al. "ImageNet: A large-scale hierarchical image database." (2009).
> [4] Esser et al. "Taming Transformers for High-Resolution Image Synthesis." (2021).

---

> ### Author Response · Authors · 2025-11-23
>
> Thank you again for your thoughtful review. We would be very grateful to know if our detailed response and the new experiments addressed your concerns. If they did, we would be deeply grateful if you would consider raising your score. Thank you for your time.

---

### Author Response · Authors · 2025-12-03
**Summary of Contributions, Revisions, and Response to Concerns**

Dear Area Chair,

Given the special circumstances of this year's review process, we have prepared a summary of our work for your convenience. It summarizes our contribution and demonstrates how our revisions address the reviewers' concerns.

$\newline$

## Our contribution

This work addresses the inefficiency of error correction in masked diffusion. While unguided re-masking (e.g., ReMDM) improves quality, it creates a computational bottleneck by requiring excessive steps (often 2-4x the sequence length). We propose Guided Star-Shaped Masked Diffusion (G-Star), a simpler, pre-trained-compatible, remasking sampler that integrates a lightweight, learnable error predictor. Instead of re-masking randomly, G-Star **guides the correction**, targeting only likely errors. This targeted refinement yields a "commanding advantage": our Pareto-front analysis (Fig. 5) confirms that 128-step G-Star provides a better quality-diversity balance than 512-step ReMDM, and is ~3x faster in wall-clock time.

$\newline$

## Reviewer's feedback

Reviewers highlighted the **importance of targeted error correction** in discrete diffusion and found our approach **well-motivated and theoretically sound** (‘sA7a’, ‘jgo2’, ‘ocUj’, ‘pM5q’). They appreciated that the star-shaped formulation **preserves MDLM’s training objective** and is **compatible with pretrained backbones** (‘pM5q’, ‘ocUj’), and that the guided error predictor is **lightweight yet effective**, with **strong gains in low-step regimes** and **across downstream tasks including code and instruction following** (‘sA7a’, ‘jgo2’, ‘ocUj’, ‘pM5q’). Building on this positive feedback, we focused on addressing the key concerns. We summarize our main updates and new experiments below.

$\newline$

### Key updates in the revision

> **1. Pareto-front analysis of quality–diversity trade-offs (‘sA7a’)**

* To move beyond point-wise metric comparisons, we added a Pareto-front analysis (Section 4.4, Figure 5) by varying the diffusion temperature.
* Across all compute budgets, this new analysis shows G-Star-loop consistently dominates the MDLM baseline and the unguided ReMDM-loop, achieving a superior quality–diversity balance.
* Notably, our 128-step G-Star-loop model achieves **both lower perplexity and higher diversity** than the best-tuned 512-step ReMDM configuration.


> **2. Clarifying novelty vs. ReMDM and Star-Shaped DDPMs (‘sA7a’, ‘jgo2’)**

* We clarified that our core novelty is two-fold: (i) a simpler Star+ formulation for remasking (Sec 3.1), and (ii) the **learnable guided predictor ($g_{\phi}$)** (Sec 3.2), which makes error correction targeted and efficient.
* This guided predictor directly addresses the core inefficiency of ReMDM, which uses unguided re-masking.
* We also distinguished our work from Okhotin et al. (Star-Shaped DDPMs). While their model requires complete model retraining and schedule tuning, a key practical advantage of our method is its compatibility with pre-trained MDLMs **without fine-tuning**, along with a targeted error predictor.


> **3. Inference overhead: wall-clock latency, NFEs, and memory (‘sA7a’, ‘jgo2’, ‘pM5q’)**

* A new Appendix D provides a **quantitative latency study** on OpenWebText (L=512, batch size 1), reporting wall-clock time, NFEs and memory overhead for MDLM, ReMDM, Star, and all G-Star variants, as well as AR baseline.
* Star and ReMDM have identical cost to MDLM.
* G-Star incurs a controlled overhead. In our main setup this yields:
   * $\approx 1.5\times$ NFEs for the strongest G-Star-loop variant.
   * A much smaller overhead ($\approx 4–20\%$) for lightweight predictors.
* Despite this, 128-step G-Star is $\approx 2–3\times$ faster in wall-clock time than 512-step ReMDM, while achieving better quality-diversity trade-off.


> **4. Presentation, clarity, and missing details (‘sA7a’, ‘jgo2’, ‘ocUj’)**

* To address concerns that the method was "conceptually dense" (ocUj), we moved the illustrative figure from the appendix to the main paper (now Figure 1). We also revised Section 3 to provide an intuitive, high-level walk-through of the Star and G-Star samplers before the algorithm.
* We extended the description of the error predictor's training (architecture, hyperparameters, compute budget for all setups) in Appendix E.
* To improve practical usability and provide guidance on hyperparameters, we added a new Appendix G. This section offers concrete recommendations and intuition for selecting key hyperparameters, such as $t_{on}$ and the denoiser/predictor temperatures.
* We also corrected all noted presentation issues, including table ordering and the typographical errors.

---

> ### Author Response · Authors · 2025-12-03
>
> > **5. Robustness, generalization, and sensitivity (‘sA7a’, ‘jgo2’, ‘ocUj’, ‘pM5q’)**
>
> * To address cross-domain robustness, we added a **new OOD analysis (Appendix C.5)**. The predictor (trained on OWT) generalized exceptionally well to four unseen domains (TinyStories, The Stack (Python code), OpenWebMath, CNN/DailyMail), maintaining a high AUC-ROC (0.90-0.98). This suggests it learns general diffusion error patterns.
> * Per Reviewer sA7a's request, we conducted a **new experiment on ImageNet**. The results (shared in our rebuttal) showed a benefit in the low-step (T=16) regime, but the overall impact was less pronounced than in text. We provided a detailed analysis of *why* this VQ-GAN setting (e.g., token interchangeability, decoder robustness) makes re-masking less impactful.
> * To investigate the **train-inference distribution shift**, we ran a **new experiment** to reduce this mismatch. This new training setup only yielded marginal gains (e.g., 57.3 $\rightarrow$ 59.7 MAUVE) while doubling the training cost. We conclude the standard training offers better practical trade-off, while more sophisticated mismatch mitigation is left for future work.
> * We clarified that the $t_{sw}$ hyperparameter was intentionally not re-tuned for G-Star. This was to demonstrate method's superiority even when using a parameter optimized for that baseline — a value that we confirmed lies within a robust effective range.
>
>
> > **6. Methodological limitations and edit expressivity (‘pM5q’)**
>
> * We agree that the need for a hybrid schedule (switching from MDLM drafting to G-Star refinement) is a limitation. However, we clarified that this is what provides a major practical advantage: the flexibility to apply G-Star to **any pre-trained** MDLM **without costly retraining**.
> * We acknowledge that the sampler is limited to in-place substitutions (no insert/delete), which restricts structural edits. We noted that it can still perform non-local edits by rewriting contiguous spans (see Appendix F). We explicitly highlighted this in the Limitations section as an important direction for future work.
>
> ---
>
> We believe our detailed responses, new experiments, and manuscript revisions fully address the key concerns raised during the review. The new results, particularly that our 128-step guided model achieves a superior quality-diversity balance than the 512-step unguided baseline, strongly reinforce the paper's core contribution and practical value. We are grateful for the reviewers' feedback, which has significantly strengthened the paper. Thank you for your time and consideration.

---

### Meta-Review · Area_Chair_HC32 · 2026-01-07

**Summary:**

The reviewers main concerns were
* Novelty vs. ReMDM
* Computational overhead
* Quality vs. diversity tradeoff

I am quite surprised and concerned about the review quality for this work. Specifically,
* No reviewer brought up the large literature on remasked discrete diffusion model sampling. Including works that are essentially identical or more general than this work.
* The "star-shaped formulation" presented in this work is the standard formulation. All recent discrete diffusion methods use $x_0$ prediction

I believe this reflects an unfamiliarity with the literature or a lack of effort from the reviewers. Given this I've decided to read this work deeply and offer my own perspective as someone familiar with this area. I regret the low quality of reviews for this work.

------

### Summary

This paper presents guided star-shaped masked diffusion which adds selective remasking to ReMDM.


### Strengths and Weaknesses

Strengths:
* Identifies a transition point during sampling where it is optimal to turn on the star shaped sampler.

Weaknesses:
* **Lack of motivation for star-shaped sampling**: Star shaped sampling seems to be a contrived problem as there are already non-star-shaped masked diffusion models which are able to do remasking. There is no empirical nor theoretical justification for this formulation that I can see.
* **Essential comparisons to related work are missing**: This paper misses a large fraction of the literature on improved samplers for masked diffusion models. Some of which are extremely similar to this work. I outline a few of the most relevant ones here:
  * Zheng et al. 2024 [1] (RDM) uses remasking based on the confidence of the denoiser, although is also not strictly start shaped.
  * DDPD [2] uses remasking based on a separate trained planner network and the planner is conditionally independent of $x_t$ given $\hat{x}_0$, this method also uses the planner for unmasking.
  * P2 [3] also uses remasking based on separate planner which is trained or derived from the denoiser.

  While I see slight differences between these works and G-Star+, in my view these works need to be discussed and compared against. Specifically, the novelty of the second main contribution is unclear with respect to these works.
* What is referred to as perplexity is not the standard measure of perplexity (which cannot be easily measured in discrete diffusion models, especially with more exotic samplers), but what is known as *generative* perplexity. This is a very different metric, and is much more susceptible to errors, specifically lack of diversity.

> Notably, this star-shaped sampling process establishes a direct connection to the ReMDM framework (Wang et al.). Specifically, our sampler is mathematically equivalent to the ReMDM sampler when its probability is set to $\sigma_t = 1 − \alpha_s$.

This is never shown. I'm not sure this true as it seems even with the proposed substitution reMDM depends on the $x_t$ through the mask state where the proposed method does not. Also I'm not sure what $\alpha_s$ is here, I would guess this is supposed to be $t$ or $t-1$? In any case, this requires further exploration.

Claim 1 seems repeated from ReMDM with no citation.

Minor details
* It would be cleaner if figure 1 used the same indexing as the rest of the paper so $t$ and $t-1$ rather than $t+1$ and $t$.
* In the preliminaries it is stated:

  > The reverse process is parameterized by a neural network, $f_\theta(x_t, t)$, which is trained to predict the probability distribution over the original data, $p_\theta(x_0 | x_t)$.

  This is incorrect, $f_\theta$ is trained to predict the **expectation** over $x_0$ given $x_t$, not the full distribution.

References:
1. Lin Zheng, Jianbo Yuan, Lei Yu, and Lingpeng Kong. A reparameterized discrete diffusion model for text generation. COLM, 2024.
2. Sulin Liu, Juno Nam, Andrew Campbell, Hannes Stärk, Yilun Xu, T. Jaakkola, and Rafael G’omez-Bombarelli. Think while you generate: Discrete diffusion with planned denoising. ICLR, 2024.
3. Fred Zhangzhi Peng, Zachary Bezemek, Sawan Patel, Jarrid Rector-Brooks, Sherwood Yao, Avishek Joey Bose, Alexander Tong, and Pranam Chatterjee. Path planning for masked diffusion model sampling, 2025

**Reviewer Concerns:**

I believe all concerns from the reviewers were adequately addressed.

**Reviewer Scores:**

sA7a 4 --> 4 I think the novelty here would be a problem if I was in this reviewer's shoes even after the rebuttal

jgo2 6 --> (6 or possibly 4): In my opinion this reviewer was initially concerned about novelty, and given that the application to error correction in masked discrete diffusion models is also not novel, this would potentially downgrade the score

ocUj 6 --> 6: the primary concerns were around clarity, I don't think it is likely that this would cause the reviewer to bump to an 8

pM5q 6 --> 8: All concerns were addressed

---

### Decision · Program_Chairs · 2026-01-26

Reject